# Monitoring the Daily Evolution and Extent of Snow Drought

Benjamin J. Hatchett[1], Alan M. Rhoades[2], and Daniel J. McEvoy[1]

[1]Desert Research Institute, Reno, Nevada, USA, 89512
[2]Climate and Ecosystem Sciences Division, Lawrence Berkeley National Laboratory, Berkeley, California, USA, 94720

**Correspondence:** Benjamin J. Hatchett (benjamin.hatchett@gmail.com)

**Abstract.** Snow droughts are commonly defined as below average snowpack at a point in time, typically 1 April in the western United States (wUS). This definition is valuable for interpreting the state of the snowpack but obscures the temporal evolution of snow drought. Borrowing from dynamical systems theory, we applied phase diagrams to visually examine the daily evolution of snow water equivalent (SWE) and accumulated precipitation conditions in maritime, intermountain, and continental snow climates in the wUS using station observations as well as spatially distributed estimates of SWE and precipitation. Using a percentile-based drought definition, phase diagrams of SWE and precipitation highlighted decision-relevant aspects of snow drought such as onset, evolution, and termination. The phase diagram approach can be used in tandem with spatially distributed estimates of daily SWE and precipitation to reveal variability in snow drought type and extent. When combined with streamflow or other environmental data, phase diagrams and spatial estimates of snow drought conditions can help inform drought monitoring and early warning systems and help link snow drought type and evolution to impacts on ecosystems, water resources, and recreation. A web tool is introduced allowing users to create real-time or historic snow drought phase diagrams.

## 1 Introduction

Snow-dominated mountains provide critical water resources to ecosystems and society (Viviroli et al., 2007; Sturm et al., 2017; Immerzeel et al., 2020), but their snowpacks are susceptible to climate warming (Beniston, 2003; Pepin et al., 2015; Rhoades et al., 2018c; Siirila-Woodburn et al., 2021). Warming impacts mountain regions in many ways including reductions in the amount of water stored in snowpack (Mote et al., 2018), earlier spring snowmelt (Kapnick and Hall, 2012; Musselman et al., 2021), and slower snowmelt (Musselman et al., 2017). Warming will increase the frequency of extreme rain-on-snow events (Musselman et al., 2018) and generally decreases the fraction of precipitation falling as snow (Lynn et al., 2020). As rain falls instead of snow, runoff could become less efficient (Berghuijs et al., 2014) as water is no longer stored in the snowpack and as warming increases atmospheric water demand (Fisher et al., 2017). However, seasonal shifts in energy availability altering plant available water storage (Barnhart et al., 2020) may counteract reduced snowmelt rates and changes in rain and snow partitioning, potentially buffering runoff changes in energy-limited (colder) environments.

Despite challenges in constraining mountain hydrologic sensitivity to warming, spring snowpack remains an important predictor of warm season runoff for environmental flows and human consumptive use in many regions Siirila-Woodburn et al. (2021). Peak snow water equivalent is projected to decline by 40-60% in the western United States (wUS) by end-century Siirila-Woodburn et al. (2021). For regions historically characterized by a seasonal snowpack, these declines will reduce

drought prediction skill (Livneh and Badger, 2020). Losses will be pronounced in lower elevation coastal snowpacks that are most "at risk" to warming (Nolin and Daly, 2006; Dierauer et al., 2019; Hatchett, 2021; Evan and Eisenman, 2021). Reductions in snowpack negatively impact wildlife habitat (Barsugli et al., 2020) and decrease opportunities for recreation and
tourism (Scott, 2006; Hatchett and Eisen, 2019; Crowley et al., 2019) in addition to downstream agricultural (Qin et al., 2020), environmental (Poff et al., 1997; Yarnell et al., 2020) and other economic impacts (Lund et al., 2018; Sturm et al., 2017). In rural mountain regions, winter recreation and tourism form pillars of the economy (Hagenstad et al., 2018).

Tracking snowpack throughout the wUS cool season (typically November-April) and identifying below-normal snow conditions known as "snow drought" (Cooper et al., 2016; Harpold et al., 2017; Hatchett and McEvoy, 2018; Huning and AghaK-
ouchak, 2020a) aids resource managers in making decisions (e.g., water allocations or the timing and magnitude of runoff) based on the state of the snowpack relative to past and forecast weather and snowpack conditions. Often, a point-in-time approach is used to assess snowpack conditions pertaining to runoff. For example, the date of 1 April is codified into many wUS water management agencies (Lynn et al., 2020) that depend on runoff from both seasonal and ephemeral snowpacks (Hatchett, 2021) to assess warm season water availability. The relation of this date to peak snowpack timing and its representatives of
the total volume of potential meltwater, however, varies by location and season (Trujillo and Molotch, 2014; Margulis et al., 2019; Musselman et al., 2021; Siirila-Woodburn et al., 2021). Hatchett and McEvoy (2018) highlight other challenges of the single point-in-time definition. Notably, they discussed that pre-1 April snow droughts can be obscured by later heavy snowfall and that anomalous melt events during warmer-than-normal conditions can create snow drought conditions not directly related to precipitation. In addition, no clear definition for snow drought onset has been provided. Hatchett and McEvoy (2018) used
80% of average SWE, whereas Harpold et al. (2017) used below normal. Depending on snowpack accumulation, which varies by widely within and between regions, snow droughts may occur across a range of percentages of normal and are not easily comparable. This motivates the use of a percentile-based approach to facilitate regional comparisons.

These challenges, and the need to communicate mountain hydroclimate conditions to varied user groups (e.g., the National Weather Service, natural resource managers and other decision makers (Marshall et al., 2020)), illustrate the need for easily-
accessible, informative data visualization approaches. Ideally, these visualizations capture the signals of interest for decision-relevant contexts and allow a user to track snowpack and precipitation evolution through the cool season or the entire water year (WY). Here we introduce the application of phase diagrams that enable visualizing how two variables co-vary through time. Specifically, we show the daily temporal co-evolution of snow water equivalent (SWE) and precipitation. We demonstrate the utility of this approach using examples across a range of spatial scales in wUS snow-dominated regions. We also highlight
intraseasonal and interannual snowpack variability, snow drought variation along an elevational and longitudinal transect, and how dry snow droughts (below-average precipitation and snowpack) versus warm snow droughts (above-average precipitation but below-average snowpack) differ. Last, a web-based tool enabling the creation of phase diagrams "in real time" based on the SNOTEL network is introduced: https://wrcc.dri.edu/my/climate/snow-drought-tracker.

## 2 Data

 ## 2.1 Station Observations

Daily observations of SWE and accumulated WY precipitation (the WY begins on 1 October and ends on 30 September, with the year corresponding to the latter date) were acquired from seven SNOwpack TELemetry (SNOTEL) stations from the Natural Resources Conservation Survey (https://www.wcc.nrcs.usda.gov/snow/) across the wUS (Figure 1; Table 1). SNOTEL is a long-term, quality-controlled, surface-based network for observing precipitation and snow in wUS mountains (Serreze et al., 1999). We used SNOTEL stations located in California, Colorado, Nevada, Utah, and Washington (Table 1). These stations capture three primary snow climates (maritime, intermountain, and continental; (Mock and Birkeland, 2000)). We acquired daily SNOTEL data spanning the period of record (typically beginning in the 1980s) through 31 May 2020 for complete WYs. We recommend only using stations with at least 20 years of data. In our example highlighting the web-based tool, we used an end date of 8 March 2021 to show the real-time application of phase diagrams. Last, we highlight an example of how snow drought conditions influence streamflow and to show how to link phase diagrams with hydrologic outcomes. To do so, we acquired daily streamflow for WYs 1943–2019 from the U.S. Geological Survey Gage 12082500, located on the unimpaired Nisqually River, near the Paradise, Washington SNOTEL site (Figure 1b).

**Table 1.** Metadata for western United States SNOwpack TELemetry (SNOTEL) stations used to generate the phase diagrams.

| Station Name | Elev. m) | Lat (°N) | Lon (°W) | Start Date | Snow Climate |
|---|---|---|---|---|---|
| CSS Lab, CA | 2201 | 39.33 | -120.37 | Oct 1983 | Maritime |
| Mill-D North, UT | 2733 | 40.66 | -111.64 | Oct 1988 | Intermountain |
| Mount Rose Ski Area, NV | 2683 | 39.32 | -119.89 | Oct 1980 | Intermountain |
| Paradise, WA | 1564 | 46.78 | -121.75 | Oct 1980 | Maritime |
| Red Mountain Pass, CO | 3414 | 37.89 | -107.71 | Oct 1980 | Continental |
| Tahoe City Cross, CA | 2072 | 39.32 | -120.15 | Oct 1980 | Maritime |
| Virginia Lakes, CA | 2866 | 38.07 | -119.23 | Oct 1978 | Intermountain |

## 2.2 Gridded Observational Products

To add a spatial component to station-based SWE and precipitation phase diagrams, we utilized daily gridded 4 km resolution estimates of SWE for the continental U.S. produced by the University of Arizona (hereafter UAswe; Zeng et al. (2018); Broxton et al. (2019)). The UAswe product spans WYs 1982–2020. Daily, gridded 4 km spatial resolution precipitation was acquired from gridMET (Abatzoglou, 2013). Phase diagrams can be applied to any long-term daily *in-situ* and/or gridded SWE product and are not limited to the observational products chosen in this study. Examples of watershed-averaged phase diagrams are presented and compared with nearby SNOTEL stations for two, eight digit U.S. Geological Survey Hydrologic Unit Codes (HUC-8) watersheds (Seaber et al., 1987) in the Sierra Nevada (The Upper Yuba River Basin and the Tuolumne River Basin).

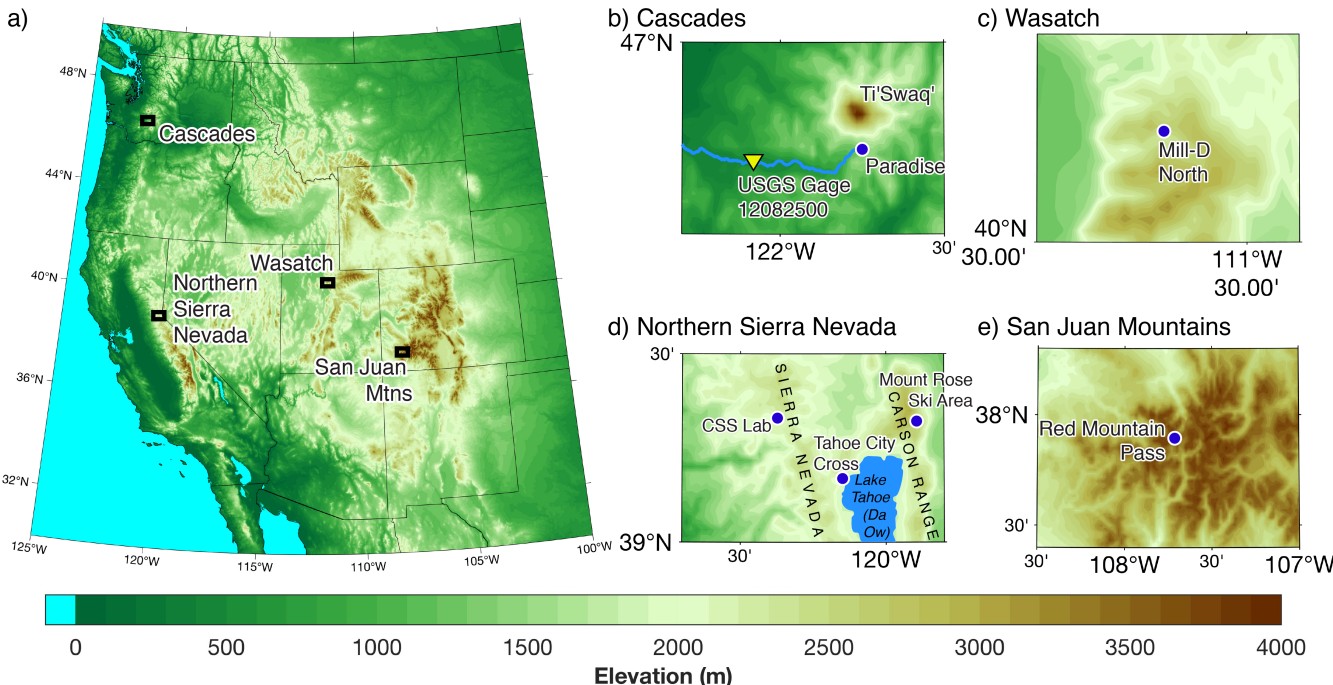

**Figure 1. (a)** Digital elevation map of western United States topography (in m) from ETOPO (Amante and Eakins, 2009) showing focus study areas: **(b)** the Cascade Mountains, **(c)** the Wasatch Mountains, **(d)** the Northern Sierra Nevada, and **(e)** the San Juan Mountains. SNOTEL stations are shown by blue dots. The yellow triangle indicates the U.S. Geological Survey Gage 12082500 on the Nisqually River.

## 3 Methods

### 3.1 Visualizing snow drought with a phase diagram

The concept of phase diagrams were initially developed by Ludwig Boltzmann, Henri Poincaré, and Josiah Willard Gibbs (Nolte, 2010). The intent was to represent all possible states of a dynamical system, such as a particle's position and momentum.

85   Many disciplines now use phase diagrams (also referred to as phase space diagrams)—including nonlinear dynamics, chaos theory, as well as statistical and quantum mechanics. Each parameter of the system in phase diagrams is represented by an axis of a multidimensional space. In a two-dimensional system, each point on the phase plane represents a combination of the system's parameters, with the evolution of the system's state through time tracing a line called the phase space trajectory. The phase space trajectory begins at the point representing the initial conditions. Depending on the application, the trajectory

90   continues indefinitely or until the time period of interest has elapsed.

Inspired by the simplicity of phase diagrams, specifically the Wheeler-Hendon phase diagrams used to track the phase and life cycle of the tropical intraseasonal Madden-Julian Oscillation (Wheeler and Hendon, 2004), our purpose is to show how this visualization approach can track SWE and precipitation conditions during the cool season. We aim to track the phase space

of cool season mountain hydroclimate conditions in order to link them to the trajectory of snow drought conditions (dry and warm; (Harpold et al., 2017)) to the hydrometeorological events (Hatchett and McEvoy, 2018) shaping the trajectories. Thus, phase diagrams can help diagnose snow drought onset, termination, duration, type, and severity as well as explore timing and characteristics of large 'drought-busting' storms (Dettinger, 2013). By implicitly including the time-varying effects of precipitation and temperature, phase diagrams provide a unique perspective over time series plots in tracking snowpack conditions. For instance, phase diagrams can clearly show the abrupt changes in one or both variables during major accumulation or melt events.

### 3.1.1 Creating the Snow Drought Phase Diagram

For each station, we calculated daily percentiles of accumulated precipitation and SWE from 1 November to 30 April (or 31 May) using a seven-day moving window centered on each calendar day to reduce seasonality effects (Montecinos et al., 2017; Shortridge et al., 2019). Results were not sensitive to moving windows sized between zero and 15 days. We calculated percentiles using the period of record. Following Huning and AghaKouchak (2020a), we used the U.S. Drought Monitor "D scale" (Svoboda et al., 2002) to characterize snow drought as abnormally dry (D0), moderate drought (D1), severe drought (D2), extreme drought (D3), and exceptional drought (D4) for values between the $30^{th}$-$20^{th}$, $20^{th}$-$10^{th}$, $10^{th}$-$5^{th}$, $5^{th}$-$2^{nd}$, and below the $2^{nd}$ percentiles, respectively. Following the Drought Monitor scale, snow drought is defined as SWE percentiles less than or equal the $30^{th}$ percentile, which is slightly more inclusive than Marshall et al. (2019) who selected the $25^{th}$ percentile as their threshold. Accumulated precipitation percentiles were plotted on the x-axis and SWE percentile on the y-axis. Each daily point was coloured by the corresponding month and connected by a line to create the phase trajectory. Snow drought severity, following the D scale, were denoted by colored lines. We defined the start of snow drought phase diagrams at the beginning of onset when snow accumulation typically occurs (1 November), however they could be started at any time deemed relevant (i.e., 1 October or 1 December). Each trajectory point is binned by a unique color for a given WY month and the first day of each month is indicated by an emboldened letter. Depending on location, we selected either 30 April or 31 May for the termination of trajectories, denoted by a gold star. By these times, most water-related decisions based upon snowpack have been made.

### 3.2 Analysis of Gridded Products

For each 4 km SWE grid cell, we calculated daily percentiles of median SWE from 1 October–31 May for WYs 1982–2020, again using a seven-day moving window. The same approach was performed for gridMET precipitation. Snow drought is defined similarly as above, when SWE is at or below the $30^{th}$ percentile. Several boundary regions (e.g., HUC-8) were used to calculate basin averages (means) for the gridded products whose grid points fell on or within the boundary.

### 3.3 Cumulative Discharge Calculations

Cumulative discharge at the Nisqually River U.S. Geological Survey stream gage was calculated for all complete WYs starting
on the first day (1 Oct). For each day until the end of the WY (30 Sept), the accumulated discharge was calculated. For each
WY, we calculated the date when 50% of the WY total accumulated discharge occurred. Median dates of 50% of WY total
discharge were calculated using the full period of record.

## 4 Results and Discussion

### 4.1 An Example Annotated Phase Diagram

WY2020 was characterized by notable snowpack and precipitation variability throughout the cool season in the northern Sierra
Nevada (Figure 2a). Both fall and late winter featured near record-low precipitation and snowpack at the Central Sierra Snow
Laboratory (CSS Lab). The upper right quadrant represents wet and snowy "Big Year" conditions when both accumulated
precipitation and SWE exceed the $50^{th}$ percentile. The upper left indicates SWE was above the median but accumulated
precipitation was below median. Trajectories into this "Dry But Snowy" quadrant can result from dry fall conditions followed
by appreciable snowfall, especially in places that normally receive fall precipitation as rain, or in lower elevation, warmer
locations when anomalous snowfall has occurred instead of rainfall. A drying fall has been identified as one signal of climatic
change in California (Luković et al., 2021) and elsewhere in the wUS (Cayan et al., 2010). This drying could induce a systematic
leftward shift in future phase diagram trajectories during the $21^{st}$ century. In addition, a shift in precipitation timing into the
colder months of the season could also drive a leftward shift (Gershunov et al., 2019).
Dry snow drought conditions (meteorological drought) are identified in the lower left (third quadrant) when SWE falls into
the D0-D4 range (i.e., less or equal to the $30^{th}$ percentile) and accumulated precipitation is below the median. We defined warm
snow drought when SWE falls to below the $30^{th}$ percentile and accumulated precipitation is greater than the median (lower
right, or fourth quadrant). An additional case of 'dry-but-warm' snow drought occurs when trajectories are in the dry quadrant
but skew towards the wetter side of a 1:1 line (dashed line in the dry snow drought quadrants). These conditions indicate drier-
than-average conditions overall but with precipitation events that did not favor snowpack accumulation. To facilitate connecting
various trajectories of phase diagrams with driving processes, the annotated figure is paired with a conceptual diagram showing
potential physical interpretations (Figure 2b).
    The start of WY2020 was characterized by bottom $3^{rd}$ percentile precipitation conditions with low (bottom $20^{th}$ percentile)
snowpack at the CSS Lab (Figure 2a). Precipitation falling as snow led to rapid improvement from snow drought into the "Dry
But Snowy" quadrant during late November into December, with precipitation recovering to near-median by mid-December.
Persistent dry conditions lasting from late December through mid-March, driven by a blocking ridge west of North Amer-
ica (Gibson et al., 2020), yielded snow drought onset in late January. Above-normal temperatures and dry conditions in late
February and early March caused snowpack declines to accelerate, reaching a minimum value in the $5^{th}$ percentile. Given that
California receives the majority of its annual precipitation between December and March, dry spells lead to declines in precip-

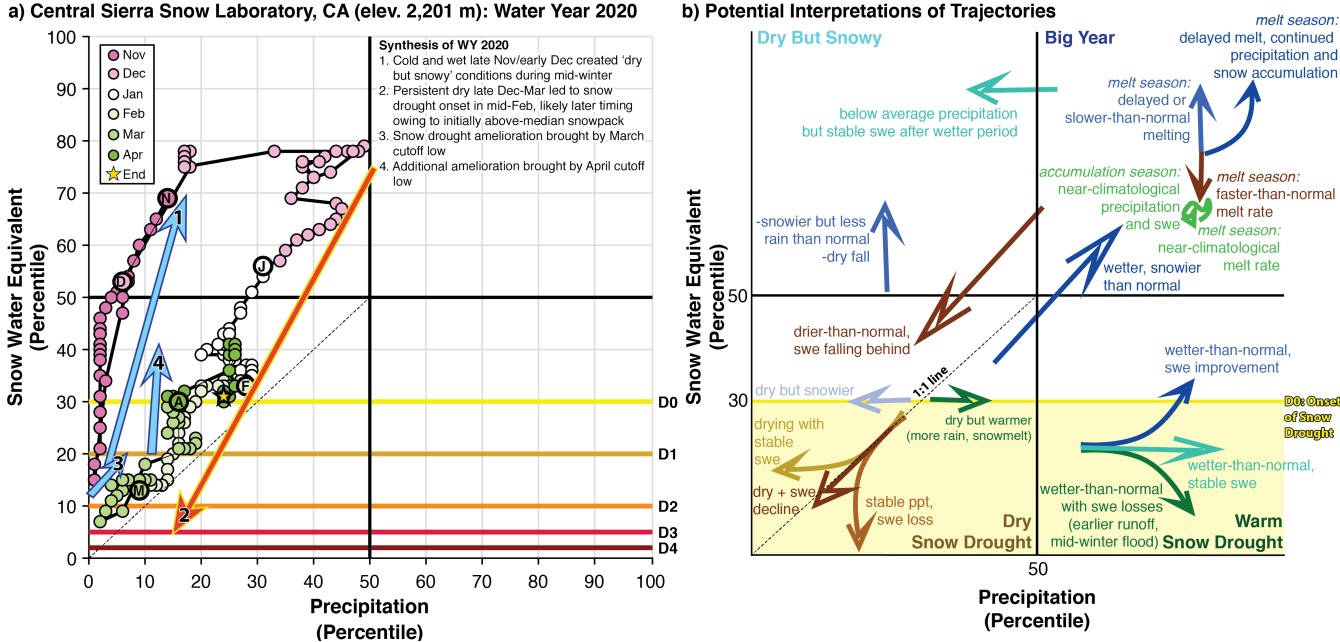

**Figure 2. (a)** Annotated phase diagram showing 1 November 2019 to 30 April 2020 at the Central Sierra Snow Laboratory (CSS Lab), California. **(b)** Conceptual phase diagram showing potential physical interpretations of seasonal evolution of various trajectories.

itation percentile (trajectories move leftward; Figure 2a). WY2020, like other Sierra Nevada drought years, was notable for its lack of atmospheric river (AR) landfalls (Hatchett et al., 2016). ARs produce abrupt upwards and/or rightwards trajectories in the phase diagram via heavy precipitation (Guan et al., 2010) enhanced by orography (Huning et al., 2017). Snow drought amelioration in late March occurred when heavy snowfall resulted from a slow-moving cutoff low pressure system (O'Hara et al., 2009). By 1 April, the historically assumed peak timing of snowpack in the wUS (e.g., Huning and AghaKouchak (2020b)),

snow drought conditions remained but SWE percentiles increased from the $5^{th}$ to nearly the $30^{th}$ percentile, though precipitation remained in the bottom $15^{th}$ percentile. Another cutoff low in early April provided additional snow that terminated snow drought conditions, however accumulated precipitation remained below the median. This highlights the importance of late spring (i.e., post-1 April) weather events in improving hydroclimatic conditions and a potential pitfall of assessing drought conditions at a single point in time. By annotating the phase diagram, the story of the cool season can be expressed to show the

key events producing observed outcomes (Figure 2b).

## 4.2    Snow Drought Variation in Time and Space

Weather events drive elevation-dependent changes in snowpack and snow drought conditions (Hatchett and McEvoy, 2018). In regions located near climatological expected rain-snow transition elevations (Jennings et al., 2018), such as the Sierra Nevada, individual storms can produce dramatically different responses in snowpack spatial variability and magnitude. ARs are a

common type of storm event yielding variable snowpack and hydrologic responses as a result of heavy precipitation with high snowline elevations (Hatchett et al., 2017; Hatchett, 2018; Henn et al., 2020) or with snowline elevations that vary widely over the duration of the storm (Lundquist et al., 2008; Hatchett et al., 2020).

     WY2018 was emblematic of the aforementioned variation in rain and snow transition elevations as both elevation- and spatially-dependent responses to storms and dry spells occurred in the Sierra Nevada (Figure 3). WY2018 began with varying

precipitation and SWE percentiles between three stations, again in the "Dry But Snowy" quadrant at the lower elevation stations (CSS Lab and Tahoe City Cross) and near climatology for the high elevation station (Mount Rose Ski Area). A late November AR event was followed by a multi-month dry spell that terminated in late February. Snowpack and precipitation increased markedly in March, (colloquially termed a "Miracle March"), due to persistent stormy conditions associated with multiple landfalling ARs associated with strong midlatitude cyclones.

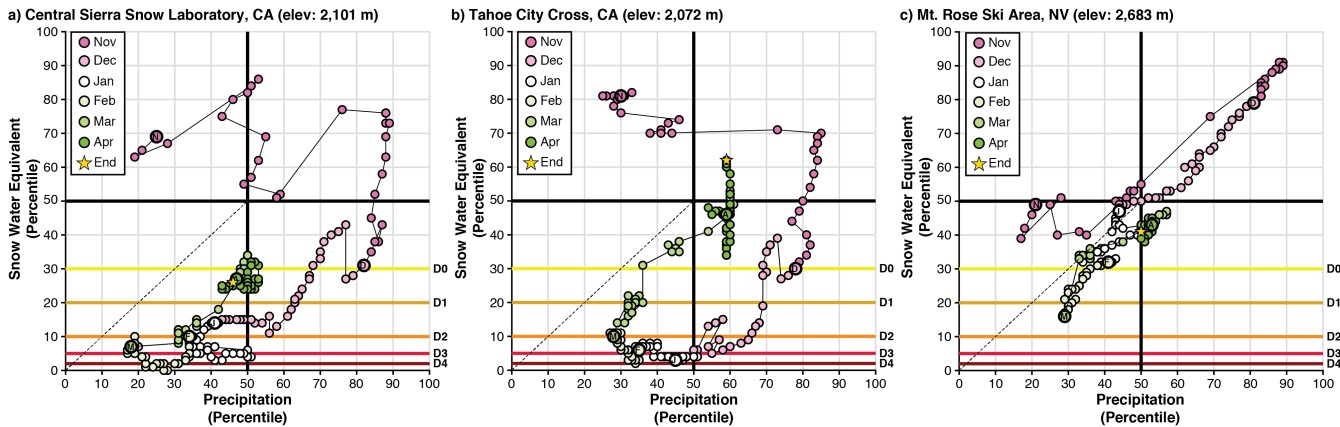

**Figure 3.** An elevation-longitudinal examination of snow drought conditions during water year (WY) 2018 in the northern Sierra Nevada of California and Nevada. Stations are ordered from west to east: **(a)** CSS Lab, **(b)** Tahoe City Cross, and **(c)** Mount Rose Ski Area.

To highlight the elevation-induced heterogeneity of snowpack response within WY2018, we investigate three different stations situated along a similar longitude (Figure 1c). The late November warm and wet storm caused the CSS Lab and Tahoe City Cross (Figure 3a-b); both maritime snow climates) to shift rightwards and then downwards into the warm snow drought quadrant because much of the precipitation fell as rain. The CSS Lab is located along the Sierra Nevada crest while Tahoe City Cross is located further east in the rain shadow of the Sierra Nevada. Unlike the other two stations, the higher elevation

Mount Rose Ski Area (hereafter "Mount Rose"), located further east in the Carson Range in a more intermountain snow climate (colder and drier than a maritime snow climate), received all snow. Mount Rose began the meteorological winter with $80^{th}$ percentile precipitation and SWE ("Big Year"; Figure 3c). The CSS Lab and Tahoe City Cross received snow early in December, briefly moving each location out of warm snow drought. During the subsequent dry spell, the lower elevation CSS Lab and Tahoe City Cross stations both moved leftward from warm snow drought into dry snow drought, with a 30 percentile

point decline in SWE through December. Dry snow drought conditions began at Mount Rose in early February. Importantly,

the role of elevation is highlighted ($\sim$600 meter range between stations) with the colder Mount Rose experiencing less dramatic snowpack declines (reaching a minimum of the $16^{th}$ percentile) compared to the warmer CSS Lab (minimum of $1^{st}$ percentile) and Tahoe City Cross (minimum of $2^{nd}$ percentile).

The return of North Pacific storminess during late February into March improved precipitation and snowpack conditions. This period also highlighted how snow drought amelioration is influenced by elevation. During this period, Mount Rose received all precipitation as snow. As a result, SWE increased by 30 percentile points (terminating snow drought) while precipitation increased from the $26^{th}$ percentile to $52^{nd}$ percentile (Figure 3c). The CSS Lab increased its SWE by 35 percentile points from the lowest on record for the date in late February to non-snow drought conditions by late March. Precipitation also increased by approximately 35 percentile points to near median values. The cold March storms demonstrated a weaker rain shadow and generally lower snowline elevations. This increased SWE at the Tahoe City Cross from the $2^{nd}$ percentile to above the $40^{th}$ percentile while precipitation also increased from the $26^{th}$ to the $60^{th}$ percentile between late February and early April (Figure 3b). Owing to "Miracle March", 1 April 2018 SWE conditions were closer to median than reflected by the majority of the winter, similar to WY2020 (Figure 2a). The record to near-record low, late winter SWE at the lower elevation CSS Lab and Tahoe City Cross are thus hidden by a single point-in-time perspective. WY2018 and WY2020 demonstrate the importance of a complete WY perspective regarding the assessment of evolving snow drought conditions, namely the importance of a few large precipitation events.

## 4.3 Warm Versus Dry Snow Drought: Implications for Runoff Timing

The warming-induced shift in precipitation phase from snow to rain has been shown in historical trends in the wUS (Lynn et al., 2020) and is projected to continue in a warmer world (Klos et al., 2014; Rhoades et al., 2018c; Musselman et al., 2018). Precipitation phase transition from snow to rain will result in more frequent warm snow droughts (Marshall et al., 2019; Huning and AghaKouchak, 2020a). This increase will disproportionately impact climatologically warmer maritime snow climates (Dierauer et al., 2019) with important implications on the headwater hydrology and downstream reservoir management strategies of these watersheds (Huang et al., 2018; Rhoades et al., 2018a; Yan et al., 2018; Rhoades et al., 2018b; Ullrich et al., 2018). To provide a comparison of WYs that experience comparable snow drought conditions with respect to snowpack percentiles, but differing drivers, we compared the WY2001 dry snow drought (Figure 4a) to the WY2015 warm snow drought (Figure 4b) at Paradise, Washington. Paradise is located in the Pacific Northwest on the south flank of Ti'Swaq' (Mount Rainier; Figure 1b). The WY2015 warm snow drought in the Pacific Northwest was a motivating and formative WY for the development of snow drought research (Cooper et al., 2016).

Paradise spent the majority of the cool season of WY2001 in the bottom $10^{th}$ precipitation percentile, a substantial difference from WY2015 when precipitation was between the $60^{th}$ and $88^{th}$ percentile between December and April. The warm snow drought resulted from an anomalous amount of precipitation largely falling as rain in the early portion of winter. Snowpack conditions marginally increased throughout WY2001 from below the $10^{th}$ percentile in February to the $20^{th}$ percentile by the end of the cool season (Figure 1a). However in WY2015, Paradise maintained fairly consistent SWE percentiles below the $10^{th}$ percentile from February to May. The leftward trajectory of precipitation during February 2015 is indicative of drier-

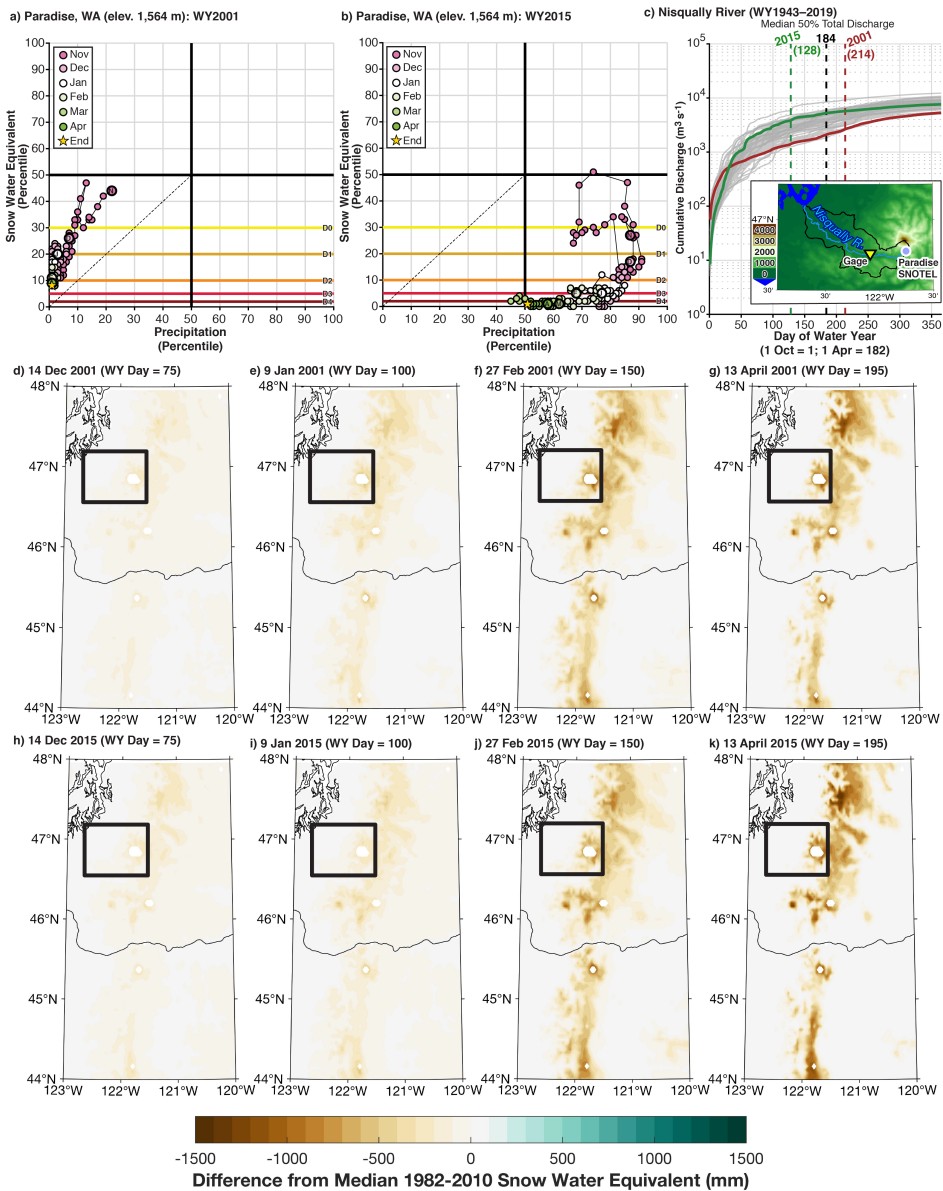

**Figure 4.** Comparison of dry **(a)** and warm **(b)** snow drought conditions in the Pacific Northwest at Paradise, Washington during water years (WY) 2001 and 2015, respectively. **(c)** Cumulative discharge (in $m^3s^{-1}$) from the Nisqually River with vertical dashed lines indicating the date at which 50% of the total WY runoff occurred. **(d–g)** Spatial snow water equivalent anomalies (in mm) during WY2001 from the UAswe product (Zeng et al., 2018). **(h–k)** As in **(d–g)** but for WY2015.

than-normal conditions followed by generally dry conditions (Figure 1b). Weak snow drought amelioration occurred in 2001,

minimizing its water resource impacts, whereas none occurred in 2015, further highlighting the importance of monitoring snow drought conditions, and type, over an entire WY.

Figure 4(a-b) shows the entire WY phase diagrams and a SWE spatial extent snapshot at various times (d-k), relative to median climatology, for WY2001 and WY2015. We also highlight the differences in hydrologic outcomes between these dry

and warm snow drought years (Figure 4c). WY2001 had the second lowest cumulative flows for the Nisqually River in the period studied (WY1943–2019), but 50% of the cumulative WY2001 flow occurred 30 days *later* than the median date (3 April) at which half the Nisqually flow occurs. In contrast, WY2015 demonstrated middle-of-the-range total WY flow ($48^{th}$ of 77 years) but achieved 50% of the WY flow 56 days *earlier* than average. This indicates a key difference between the WYs. WY2015 had less snow water stored later into the season than WY2001, markedly influencing summer streamflow. During

both seasons, despite the vastly different precipitation regimes, spatial SWE anomalies are not markedly different during mid-December (Figures 4d and 4h), mid-January (Figures 4e and 4i), or late February (Figures 4f and 4j). Consistent with lower SWE percentiles at Paradise during WY2015 compared to WY2001, as shown on the phase diagrams, SWE anomalies are modestly more negative. The lack of mountain snowpack during WY2015 was more notable than WY2001 (Figures 3g and 4k). The comparatively better spring snowpack in WY2001 likely helped maintain flows later into the year despite an otherwise

dry year.

## 4.4 Interannual Variability

Enhanced interannual precipitation variability and decreased peak SWE variability are expectations of a warming climate (Boer, 2009; Pendergrass et al., 2017; Marshall et al., 2019). Comparisons of extreme years and their outcomes providing valuable object lessons for water managers and other resource planners (Hossain et al., 2015; Sterle et al., 2019). Phase diagrams

allow for direct comparisons between WYs, helping to identify key differences in hydroclimate variability for a particular region of interest. For example, Red Mountain Pass, located in a high elevation, continental snow climate within the San Juan Mountains of southwestern Colorado, is used to compare two late cool season outcomes that represent two hydroclimatic extremes. The majority of WY2011 showed phase trajectories in the 'Big Year' first quadrant (Figure 5a) after a slightly below-average start to snowpack totals between October and early December. A stormy December increased SWE and precipitation

percentiles. Stormy weather continued in April and May, preventing snowmelt and causing precipitation and SWE percentiles to increase. WY2012 began with above-average precipitation and snowpack in fall but drier-than-normal conditions throughout winter which resulted in dry snow drought onset in December (Figure 5b). Modest snow drought amelioration occurred in early March, but with a few exceptions in April, dry conditions persisted through May. This led to the onset of dry snow drought, again, via rapid snowmelt and below-average precipitation.

Spatial SWE distributions are consistent with the phase diagrams (Figure 5c-j). In both years, SWE anomalies increased throughout the accumulation season and then accelerated in late spring. Compared to the emerging drought signal during WY2012, WY2011 did not demonstrate widespread positive SWE anomalies throughout the year. Between January and April, lower elevation regions experienced below-normal SWE anomalies (Figure 5c-d), whereas higher elevations had above-normal

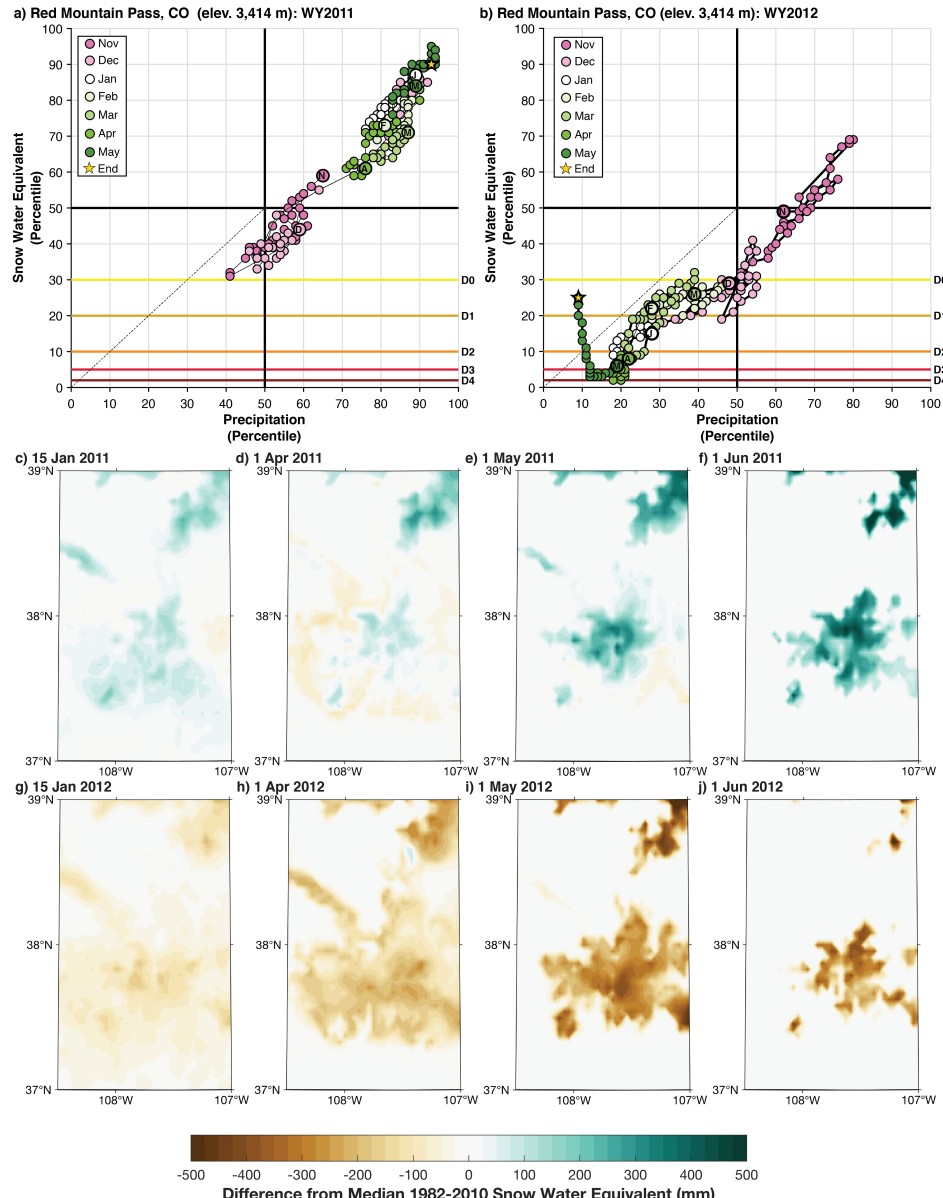

**Figure 5.** Comparison of an anomalously snowy and wet "big year" **(a)** and anomalously dry year **(b)** in the San Juan Mountains at Red Mountain, Colorado during water years (WY) 2011 and 2012, respectively. **(c-f)** Spatial snow water equivalent anomalies (in mm) during WY2011 for midwinter, early, middle, and late spring from the UAswe product (Zeng et al., 2018). **(g-j)** As in **(c-f)** but for WY2012.

SWE. This difference resulted from above-normal temperatures and below-normal precipitation, likely driven by snow-albedo
feedbacks that were enhanced at lower elevations where snowmelt occurred (Groisman et al., 1994; Stieglitz et al., 2003).

## 4.5 Towards Visualizing the Type and Extent of Snow Drought Across Space

We applied SNOTEL station data to create the phase diagrams, but a challenge in mountain environments is the lack of reliable, well-distributed, long-term observations. In lieu of station data, gridded observational products commonly inform natural resource decision-making and research efforts. The necessary components exist to create phase diagrams using gridded meteorological products (Daly et al., 2008; Abatzoglou, 2013), observation-based snow datasets (Zeng et al., 2018; Margulis et al., 2016), and output from hydrological simulations (Livneh et al., 2015). The challenge is how to aggregate spatial information to extract maximum information value for the application of interest (e.g., water management, avalanche forecasting, ecosystem processes) regarding the state and evolution of snowpack conditions at relevant scales (e.g., the full watershed, a sub-basin, or within specific elevation bands) in complex terrain. We have performed a first step towards this goal. Initial methods to broaden the approach could be performed by: (1) binning regions by similar elevation, watershed, slope, aspect, and/or land cover type; (2) identifying areas that co-vary together in time and space using techniques such as principal component or cluster analysis; and (3) subjective grouping based on anecdotal information from managers. Creating meaningful phase diagrams for varied management and scientific applications using spatially distributed information is the primary goal of our ongoing research. This will allow evaluation of snow drought in regions without long-term snow-observing networks such as in the northeastern U.S. or other high mountain areas worldwide. Towards this end, we next provide examples of how spatially distributed products can visualize snow drought.

### 4.5.1 Basin-Averaged Snow Drought Phase Diagrams

Aggregating spatially distributed information to the HUC-8 scale allows the creation of phase diagrams where few or no *in situ* observations exist. If such observations do exist, watershed-aggregated phase diagrams can be compared against station data, as shown in Figure 6 for WY2020 (see Section 4.1). We examine two Sierra Nevada watersheds, the relatively low elevation Upper Yuba River Basin and the relatively high elevation Tuolumne River Basin. Both have nearby SNOTEL stations, the CSS Lab station sits at the headwaters of the Yuba River while the Virginia Lakes station is located on the lee of the Sierra Nevada crest northeast of the Tuolumne River Basin.

In both cases, similarities exist between the SNOTEL and watershed-aggregated phase diagrams (Figure 6). The SNOTEL stations, which are located at higher elevations in the watershed, show wetter (above median) and snowier (above median) early season conditions during October and November (Figure 6a,c) whereas the watersheds show below median SWE and precipitation (Figure 6b,d). Late November and December brought substantial SWE improvement, with the Upper Yuba Basin moving into the "Dry But Snowy" quadrant (Figure 6b) and the Tuolumne River Basin extending further rightwards into the "Big Year" quadrant (Figure 6d). Virginia Lakes also increased into the "Dry But Snowy" quadrant (Figure 6c). Both regions followed similar trajectories downwards and to the left (SWE and precipitation falling behind; (Figure 2b)) during the extremely dry period spanning late December into mid-March and then underwent modest SWE recoveries with the stormy spring ("Miracle March").

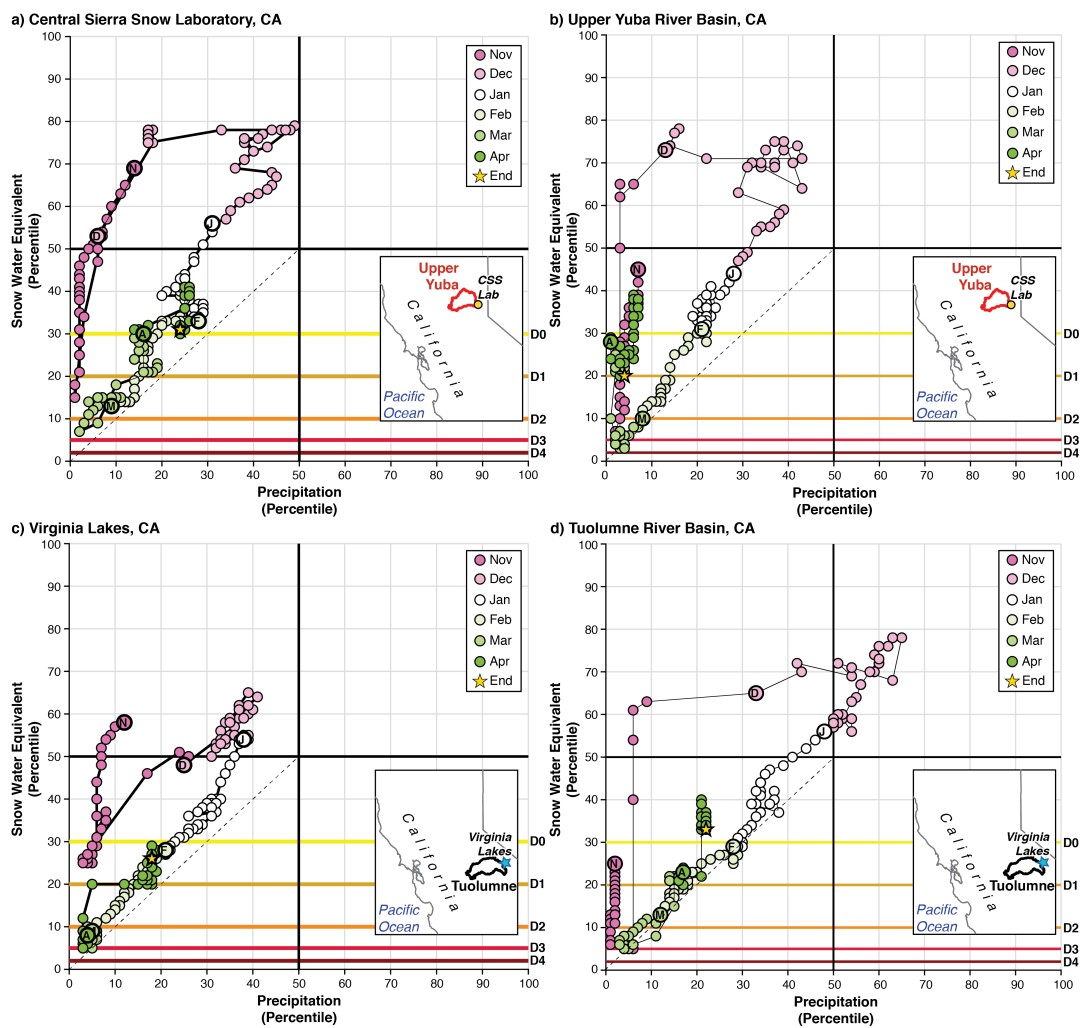

**Figure 6.** Phase diagrams for water year 2020: (**a**) CSS Lab SNOTEL, (**b**) Upper Yuba River Basin HUC-8, (**c**) Virginia Lakes SNOTEL, and (**d**) Tuolumne River Basin HUC-8. Locations of watersheds and SNOTEL stations are noted on inset maps.

By the end of the cool season (30 April), both regions showed similar SWE percentiles (within 10 points). Precipitation percentiles were different by only a few points in the Tuolumne but by over 20 points in the Upper Yuba. These differences, as well as the differences in trajectories throughout the winter, likely reflect the signal of orographically enhanced precipitation- or rain shadowing in the case of Virginia Lakes, which lies on the leeside of the Sierra Nevada crest. The CSS Lab was substantially wetter compared to the Upper Yuba during March and April of WY2020, whereas Virginia Lakes showed a similar trajectory but slightly shifted towards drier percentiles. The SNOTEL stations, likely by virtue of their location compared to the watershed hypsometry, may report higher SWE percentiles compared to the basin average if the SNOTEL station is located

at a high elevation (CSS Lab) or lower SWE percentiles if the station is found at a middle elevation on the leeward (dry) side (Virginia Lakes).

The effects of topograhy could be exacerbated at the basin scale by the inclusion of lower elevation terrain whose snow rapidly melts out during drought years (potentially accelerated by snow-albedo feedbacks, e.g., Groisman et al. (1994)). While SNOTEL stations are placed with the intention of being representative for water resources given siting restrictions, the limi-

tations created by varying topography and snowpack accumulation patterns warrants care in extrapolating data from a station to a basin or averaging across a basin in the absence of station data. Despite these differences stemming from the challenges of using limited in-situ observations, the basin-aggregated phase diagrams appear reasonably representative at capturing the broader hydroclimate conditions interpreted from phase trajectories.

### 4.5.2  Basin-Averaged Phase Diagrams for an Observation-Limited Region

The Susquehanna River Basin of the northeastern U.S. includes both seasonal and ephemeral snowpacks (Hatchett (2021); Figure 7a) that occur during extended winter (December-March; Figure 7b). Although snowpack in the Susquehanna Basin is not as critical for downstream water availability as in the western U.S., it supports winter recreation and is a critical component for ecosystems function in northern forests (Contosta et al., 2019). The Susquehanna is highly susceptible to flooding, especially during snowmelt or rain-on-snow events that contribute to enhanced runoff (Yarnal et al., 1997). Even in locations without

long-term snowpack monitoring data, basin-averaged phase diagrams can be applied to highlight a range of drought outcomes. Four examples from the Susquehanna River Basin follow.

WY1988 demonstrates a dry winter ($< 25^{th}$ percentile precipitation) that oscillated between moderate (D1) snow drought and "Dry but Snowy" (up to $67^{th}$ percentile SWE) conditions (Figure 7c). While WY1994 started off the winter season in moderate (D1) warm snow drought conditions, substantial improvement occurred from mid-December onward with consis-

tently cold conditions and a wetter-than-average February (Figure 7d) bringing a near-record snowpack late spring snowpack (Figure 7b). The snowpack and saturated soils developed during the "Big Year" of WY1994 culminated in a large flood event in early April (Marosi and Pryor, 2000). A similar situation occurred in WY1996 (not shown) with extremely wet and snowy conditions leading up to a warm, windy, and wet January storm that coincided with saturated as well as frozen soils that contributed to overland flow and widespread flooding (Yarnal et al., 1997). WY1995 demonstrates another dry snow drought

outcome, though with more severe snow drought conditions (as low as exceptional (D4)) but a slightly wetter overall winter than WY1988 (Figure 7e).

In 2016, a notable warm snow drought year occurred in the Northeastern U.S. with record temperatures (Sweet et al., 2017) and closer-to-average winter precipitation leading to dry and warm snow drought (trajectories plotting to the right of the 1:1 line; Figure 7f). The exceptionally low snowfall (much of the winter ranged from D1 to D4) exacerbated drought conditions

during the following drier-than-normal warm season, leading to 30%-70% crop losses (Sweet et al., 2017). The Susquehanna examples further indicate gridded products can be applied to create meaningful phase diagrams at the basin scale to track snow drought through time.

### 4.5.3 Snapshots from Water Year 2015 Across the Western U.S.

Using WY2015 as an example, gridded SWE and precipitation allow visualizing the spatial extent and type of snow drought
across space. Peak warm snow drought conditions in the Pacific Northwest were occurring by 1 January 2015 (Figure 8a),
consistent with the Paradise SNOTEL phase diagram (Figure 4b). At this time, much of the Intermountain West and Rocky
Mountain regions had near-average or above average (percentiles greater than the $50^{th}$, represented in purple), while several
ranges in the southern tier of the wUS (e.g., California's Sierra Nevada, the southern Basin and Range, and the Uinta Mountains
in Utah) were experiencing dry snow drought conditions. By 1 February 2015 (Figure 8b) an expansion of areas experiencing
dry snow drought occurred throughout the central and southern Rocky Mountains. Warm snow drought had started to transition
to dry snow drought in the Pacific Northwest. Dry conditions continued through February (Figure 8c). On 1 April 2015, nearly
all mountain regions were experiencing snow drought conditions (Figure 8d). Exceptions include the far northern Rockies, a
few small areas in the Colorado Rockies, and the far northern Cascades.

The transition to dry snow drought in the Pacific Northwest (Figure 8d) was also observed at the Paradise SNOTEL (Figure
4b). While the hydrologic outcome of the early winter warm snow drought included earlier runoff timing resulting from
more frequent mid-winter runoff following rain-on-snow and rain instead of snow (Hatchett and McEvoy (2018); Figure
4c), the 1 April 2015 conditions indicate dry snow drought both spatially (Figure 8d) and at the station level (Figure 4b).
This demonstrates the value of tracking snow drought and precipitation through time, as following the temporal evolution of
hydroclimate allows outcomes (e.g., runoff and spatial patterns of snowpack anomalies) to be explained with more nuance.
Such explanation is important as similar end-of-season SWE anomalies in space (compare Figures 4g, 4k, and 8d) show
different hydrologic outcomes (Figure 4c). Last, Figure 8e highlights an example of the differing elevational response of snow
drought in the Sierra Nevada (percentiles increase with increasing elevation) and the Uinta Mountains (percentiles decrease
with increasing elevation) for the same midwinter time. This example shows how sub-seasonal snowpack heterogeneity could
create differing melt-season responses (i.e., earlier snow loss at lower elevations with increasing radiation and springtime
warming) or ongoing avalanche hazards (i.e., higher elevation snowpacks are more prone to weakening when shallow).

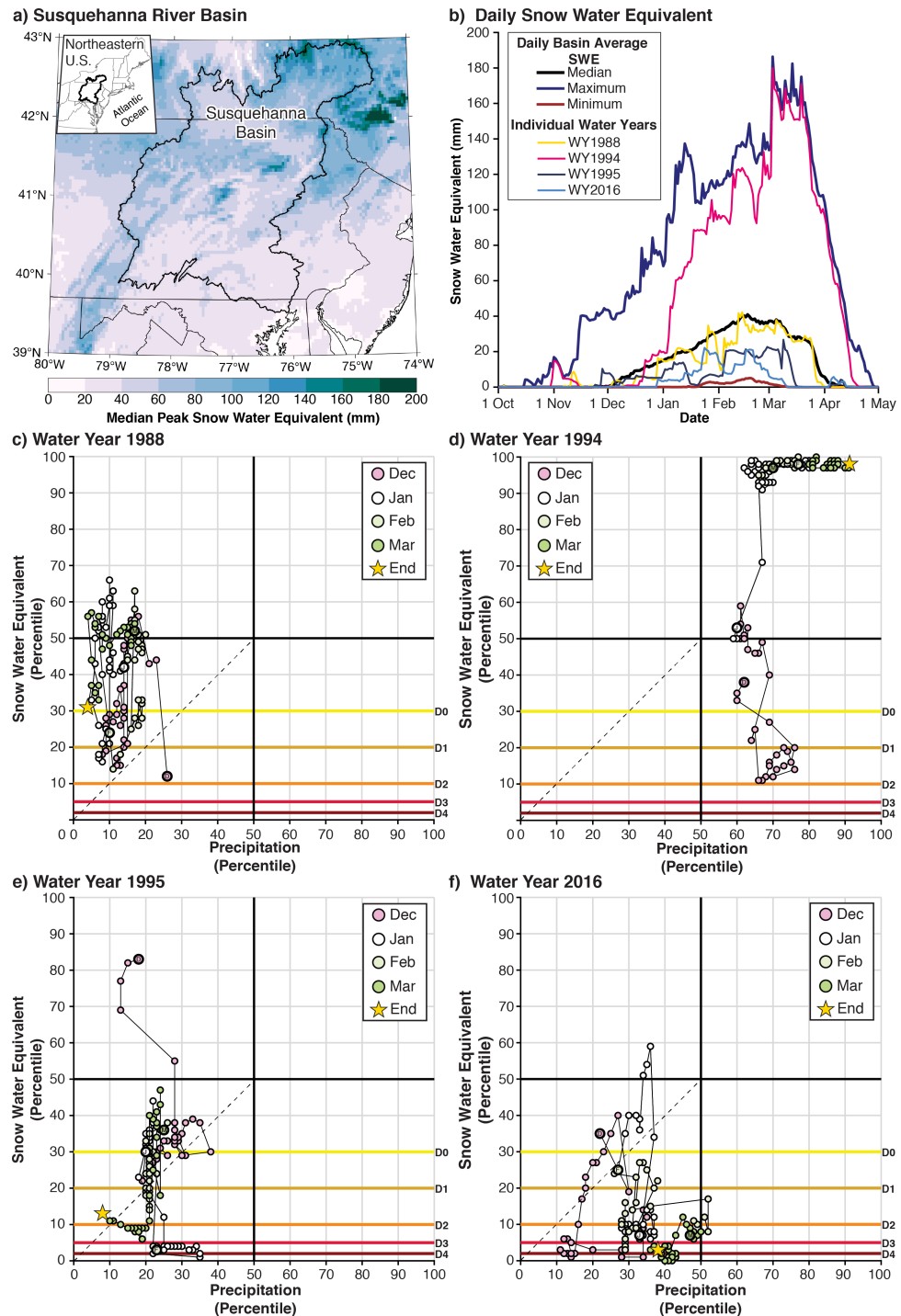

**Figure 7.** (**a**) Peak snow water equivalent (SWE) in the Susquehanna River Basin (**b**). Daily SWE climatology and time series of four water years. Phase diagrams for (**c**) Water Year (WY) 1988, (**d**) WY 1994, (**e**) WY 1995, and (**f**) WY 2016.

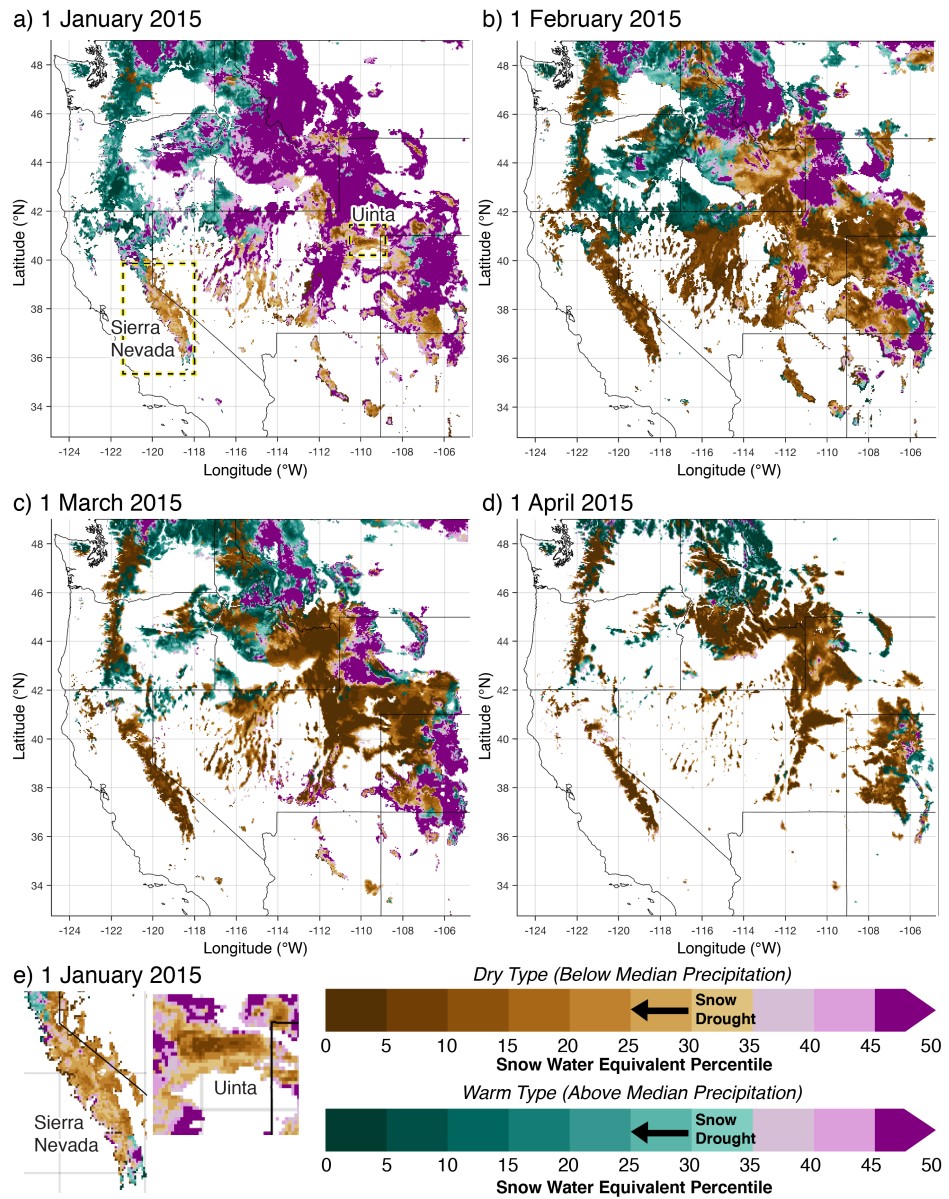

**Figure 8.** Spatial extent of snow drought conditions across the western United States on: **(a)** 1 January 2015, **(b)** 1 February 2015, **(c)** 1 March 2015, and **(d)** 1 April 2015. **(e)** The differing snow drought patterns across elevation in the Sierra Nevada (percentiles increase with increasing elevation) and the Uinta Mountains (percentiles decrease with increasing elevation). For clarity, only grid cells observing seasonal snowpacks (Hatchett, 2021) are shown.

## 4.6    Web-based Snow Drought Tracker Description

A snow drought tracker web application has been developed to provide users access to snow drought phase diagrams. The beta version is available at: https://wrcc.dri.edu/my/climate/snow-drought-tracker and simply requires users to set up a username/password to access it. The web tool is updated in near real-time across the wUS and Alaska. The SNOTEL network (Serreze et al., 1999) is the backbone of the tool with over 800 stations providing daily SWE, snow depth, precipitation, and temperature data. An interactive map (Figure 9a-b) allows for station selection via a graphical user interface. Once a station is selected, current year observations are displayed on the "Dashboard" by default (Figure 9c-d). As an example, the Mill-D North station in the Wasatch Mountains of northern Utah is shown (Figure 1b). The "Dashboard" will also display the most recent daily updated snow drought phase diagram (starting on October 1st of the WY2021; Figure 9e. The "Almanac" has several tabs showing daily SWE, snow depth, and precipitation absolute values, and percent of average SWE (Figure 9c). Note that because percent of average is more commonly used by managers, the first iteration of the tool uses percent of average (and the 80% threshold used by Hatchett and McEvoy (2018)) instead of percentiles and Drought Monitor thresholds as otherwise focused on in this manuscript. From the "Almanac" the month-to-date, calendar year-to-date, and WY-to-date precipitation values and percent of normal can be viewed (Figure 9d). In addition to the current year data found on the dashboard, historical data and graphics can be generated. Phase diagrams can be created for any year in the station record and daily time series plots can be generated for SWE, snow depth, precipitation, and temperature (e.g., Figure 9e). Figures can be downloaded as PNG or SVG files and historical data can be downloaded in CSV format. Beta testing of the snow drought tracker is being conducted by stakeholders including the National Weather Service, California Department of Water Resources, and state climatologists around the wUS. Other agencies will be encouraged to test the tool after the first round of testing and updates have concluded. Feedback from the testing will be incorporated into future upgrades of the snow drought tracker, with the goal of further developing a web-based product that provides a science-to-service-to-practice interface (Jacobs and Street, 2020). Some known limitations and gaps in the current version of the tool include the lack of spatial snow drought information (i.e., the river basin composites (Figures 6, 7), and 8 created using gridded products)) and the need to incorporate elevation gradients into snow drought monitoring.

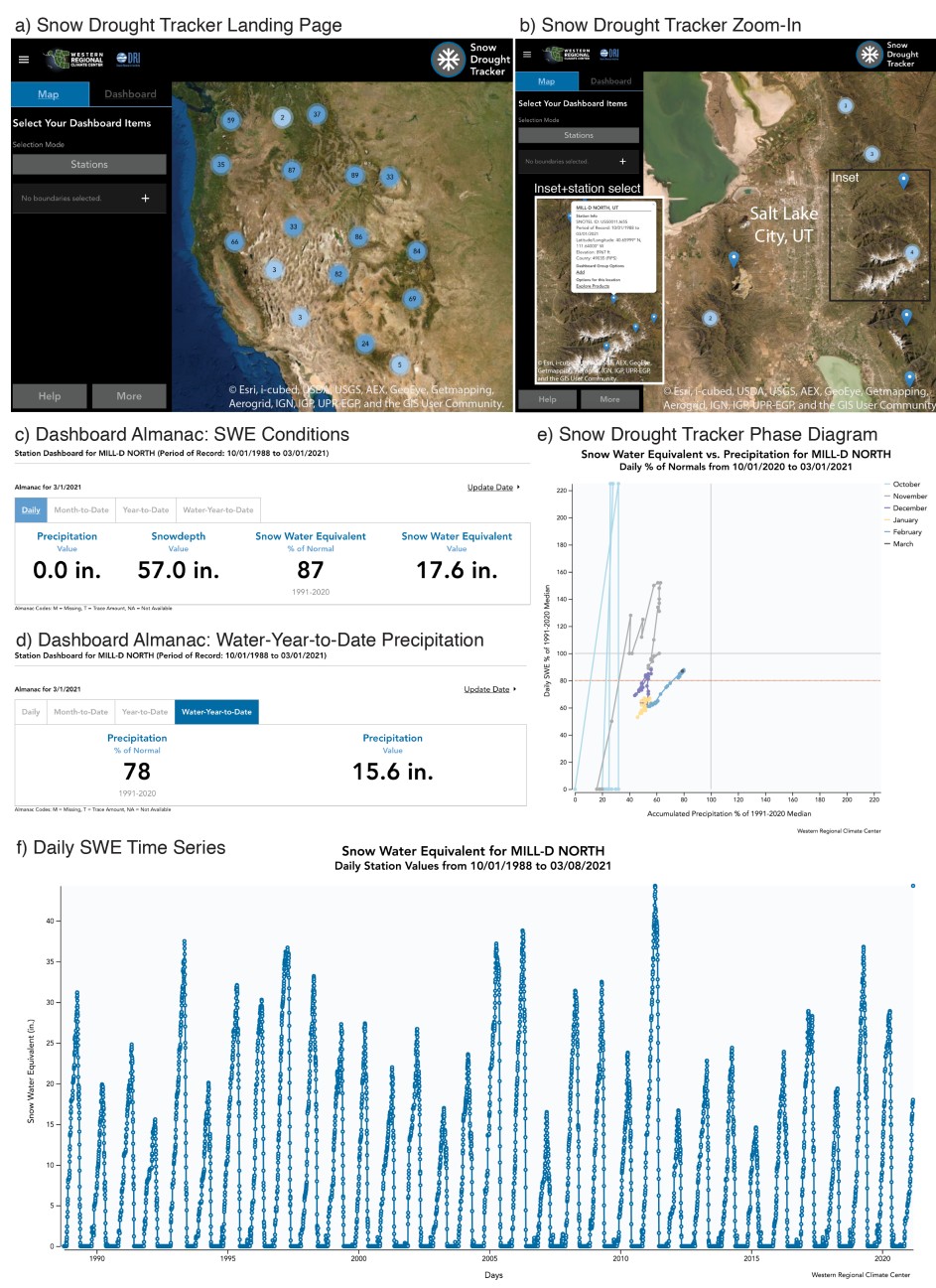

**Figure 9.** Screenshots from the beta Western Regional Climate Center (WRCC)'s Snow Drought Tracker. **(a)** Dashboard landing page. **(b)** Example screenshot assessing a station in a region of interest (inset). **(c)** Daily information tab from the Almanac. **(d)** Water-Year-to-Date Almanac tab. **(e)** Real-time phase diagram for water year 2021. **(f)** Daily snow water equivalent (SWE) time series for the period of record. Basemaps in **(a)** and **(b)** utilize ArcGIS® Online World Imagery by Esri. ArcGIS® is the intellectual property of Esri and are used herein under license. Copyright ©Esri. All rights reserved. For more information about Esri® software, please visit www.esri.com.

## 4.7 Limitations

### 4.7.1 Phase Diagrams

Our snow drought phase diagram visualization approach is not without limitations. First, we used the full period of record available for stations to calculate percentiles. In cases where stations being compared have sufficient data (i.e., at least 20 years) but differing periods of record, selecting commonly overlapping periods from which to calculate percentiles may avoid biases created by a station whose full record captures anomalous conditions (e.g., a notable wet, dry, warm, or cold set of years) compared to a station with a shorter period of record. Second, by failing to include additional environmental controls on snowpack, such as temperature, radiation, and relative humidity, phase diagrams cannot tell a complete story of the drivers of snow accumulation, ablation, and/or melt. For example, the signal of a rain-on-snow event (McCabe et al., 2007) was captured in the snow drought phase diagrams for Tahoe City Cross (see Figure 3) for an anomalously warm April AR (Hatchett, 2018). This event increased precipitation percentiles though SWE percentiles remained constant. However, when a rain-on-snow event increases net SWE, the phase diagram will not explicitly differentiate this from a snow accumulation event. Dry periods have differing snowpack outcomes during both the accumulation and ablation season depending on temperature (Hatchett and McEvoy, 2018; Xu et al., 2019) as well as how the snowpack energy budget is influenced by the deposition of dust or other light-absorbing particles on snow (Skiles and Painter, 2016; Skiles et al., 2018), cloud cover (Sumargo and Cayan, 2018), and atmospheric moisture (Harpold and Brooks, 2018). How best to include these additional parameters that help to describes changes in the phase diagram trajectories is an area of future research. Collaborations with natural resource managers, practitioners, and decision makers will be instrumental in the development of locally- or regionally-specific snow drought thresholds. Ideally, such collaborations will facilitate the use of phase diagrams for monitoring efforts (at sub-seasonal-to-inter-annual scales) and be used to evaluate past hydroclimatic extremes to improve real-time water supply monitoring (Sterle et al., 2019).

In addition to the development of locally-relevant thresholds, the identification of watershed-level sensitivity of streamflow to snow drought type and magnitude can add additional insight to drought monitoring, early warning, and how continued warming will influence mountain hydrology. Many western U.S. watersheds are characterized by dry summer conditions, relying upon snowmelt-derived groundwater recharge to maintain warm season streamflow (e.g., (Godsey et al., 2014). However, exceptions occur in humid summer climates where snowpack plays a lesser role in controlling summer hydrologic regimes, i.e., summer flow is less dependent on peak SWE (Jenicek et al., 2016). This may be the case in lower elevation watersheds in the Pacific Northwest or in regions impacted by the North American Monsoon, a warm-season precipitation regime that Carroll et al. (2020) found augments low elevation evapotranspiration and high elevation streamflow. However, the 2016 snow drought in the Northeastern U.S. (Figure 7f) shows humid climates may experience water availability challenges with reduced snowpack compounding dry and hot conditions during the warm season (Sweet et al., 2017). Watershed-specific hydroclimatic analyses could help identify and rank the risks snow drought poses to water availability and could help inform adaptation strategies to offset reductions in water availability from projected snowpack decline Siirila-Woodburn et al. (2021).

#### 4.7.2    Percentiles Versus Percent of Median: The Wilson Glade, UT avalanche incident

At present, phase diagrams on the WRCC Snow Drought Tracker show percent of median, rather than percentiles. Percent of median is commonly used in snow drought communication by the National Integrated Drought Information System's Snow Drought Tracker website (https://www.drought.gov/topics/snow-drought). A difficulty with using a percent of average rather than a percentile-based approach is it does not reflect the distribution of conditions. Percentile-based approaches provide more direct information about a given date's conditions with respect to the range of previously experienced conditions. To highlight the value of a percentile-based approach and how phase diagrams can be employed to assess the evolution of snowpack stability, an example from a 2021 avalanche incident in the Wasatch Mountains is explored.

On 11 February 2021, a skier triggered failure of a deep persistent slab R4-D2.5 avalanche event (R4 implies a large runout relative to path size, D2.5 indicates sufficiently large to bury, injure, or kill a person; Birkeland and Green (2011)) occurred on a northeast aspect at approximately 2900 m in the Wilson Glade region of the central Wasatch Mountains. Despite heroic rescue efforts, four fatalities resulted. A detailed report is available from the Utah Avalanche Center (UAC): https://utahavalanchecenter.org/avalanche/59084. The UAC's avalanche advisory for 11 February was high, indicating large human-triggered avalanches are very likely.

The phase diagram for the Mill-D North SNOTEL (Figure 10a; 3.3 km southeast of the accident site) shows early season November snowfall was followed by prolonged dry conditions into the winter months (downward and leftward trajectory), with D0 snow drought onset in early December (Figure 10a and (yellow arrow in Figure 10b)). Note snow drought onset occurs, perhaps non-intuitively, at approximately 85% of median snowpack. This indicates limited variability in SWE at this time and station: large deviations from the median value are relatively infrequent. The late November and early December dry spells led to snowpack accumulation falling behind the climatological average (Figure 10b). While some accumulation occurred during mid-December into early January, the rate of accumulation was less than climatology (Figure 10b), leading to a continued decline into D1 snow drought (Figure 10a). Percent of median SWE hovered around 65% leading up to the avalanche (Figure 10b), though SWE percentiles approached the D2 category. A transition to stormier weather in late January into February followed with gains in SWE that mirrored climatology with little change in SWE percentile or percent of median (Figure 10b). The presence of a shallow snowpack during the dry, low radiation periods in November and December promoted the formation of a persistent weak layer with striated, 3-6 mm faceted grains buried 90 cm deep in the snowpack. According to the UAC, this is the layer where failure occurred on 11 February.

The snowpack conditions leading up to the Wilson Glade avalanche show the potential disconnect between percent of median and percentile. Prior to the loading events, the percent of median values (65%) between December and early February do not directly convey the infrequency of these values as percentiles can. Percentiles show that such conditions occurred only 10-20% of the time. The user's familiarity with a location will govern the meaningfulness of percent of medians through prior experience. On the other hand, percentiles provide perspective for less-familiar users to understand the state of the snowpack. Percentiles also allow comparisons between locations in terms of snow drought severity. By recognizing both as valuable, the

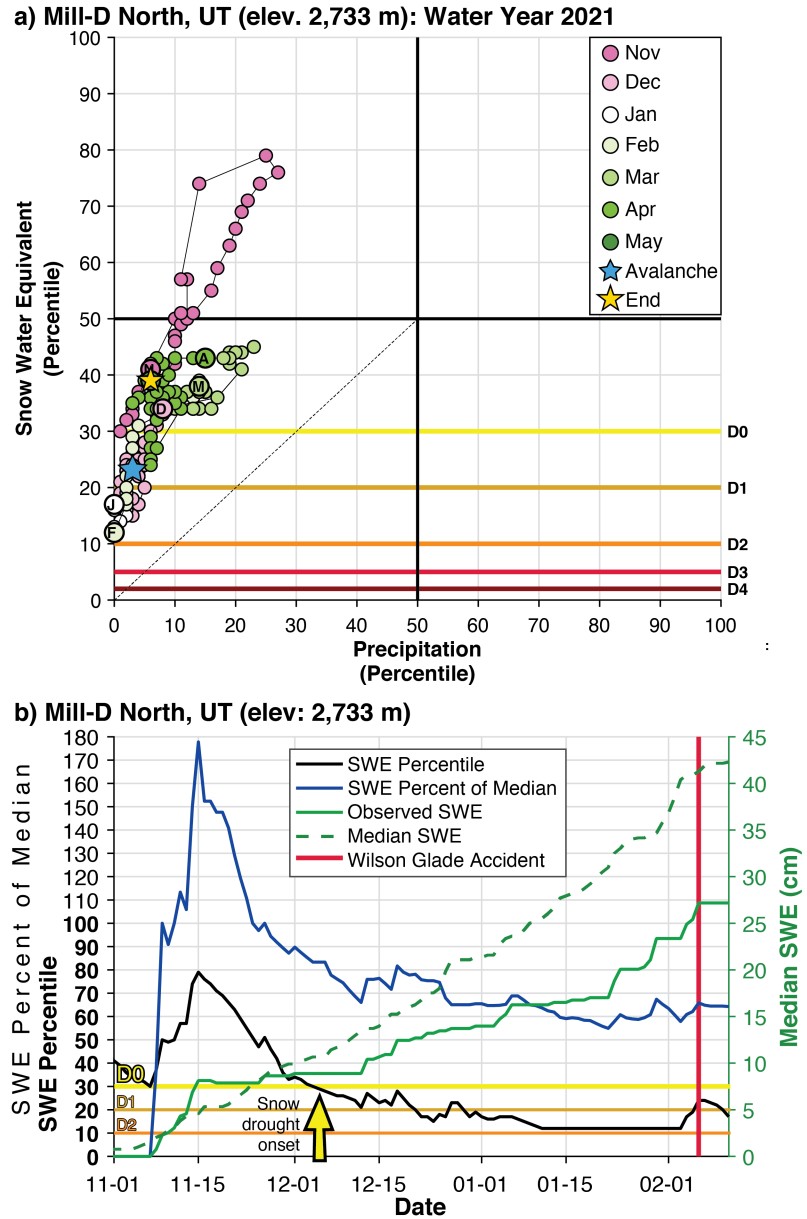

**Figure 10. (a)** Phase diagram for Mill-D North SNOTEL in the Wasatch Mountains of Utah for WY2021. **(b).** Time series of snow water equivalent (SWE) percentile (bold; left y-axis) and percent of median (left y-axis) and observed and median SWE (cm; dark blue and dashed green lines, respectively; right y-axis). The vertical red line shows the date of the Wilson Glade Avalanche accident. The horizontal gold line shows the percentile associated with the onset of snow drought conditions (D0).

option to view either on the snow drought tracker webtool is a planned improvement. Last, we recommend incorporation of percentiles into accident write-ups, such as provided by the UAC, to give this additional statistical perspective.

## 5   Conclusions

Our primary goal was to demonstrate a visualization approach to show the temporal evolution of snow drought conditions, and more broadly mountain hydroclimatic conditions, through the cool season. When annotated, phase diagrams help "tell the story" of a snow season and can help communicate the weather and climate events that shaped the outcome of peak snowpack and lifecycle of the snowpack. We provided examples showing a range of applications in various snow climates for extreme years and how additional data such as spatially distributed SWE and precipitation as well as river discharge can further enhance the utility of information provided by phase diagrams. The spatial snow drought maps and basin-aggregated phase diagrams generated using gridded data products demonstrate an approach evaluating snow drought patterns across the landscape or in sparsely observed regions.

Our approach can be extended beyond addressing the noted limitations. While our primary purpose was to show the evolution of conditions in the current year, phase diagrams are easily produced for all previous years to allow comparisons of trajectories at seasonal or monthly timescales. These diagrams can incorporate forecasts of precipitation and SWE to show how snow drought conditions may evolve at subseasonal-to-seasonal timescales. For example, inclusion of bias corrected ensembles of medium range to subseasonal forecasts of precipitation and SWE from various forecasting center model(s) can create an ensemble of plausible trajectories (or cone of uncertainty) that would provide a probabilistic perspective to explore snow-drought evolution. They can also be applied to investigate how climate change may alter phase diagram trajectories and/or residence times of WY snowpack conditions in particular quadrants of the phase diagram. However, as warming shifts the distribution of early, late, and spring peak snowpack towards lower values, it is worth considering holding the historic period from which percentiles are calculated constant over a management-relevant time period. This would reflect the historic conditions from which water management made assumptions about snowpack and water availability (Siirila-Woodburn et al., 2021).

Ultimately, phase diagrams could become useful tools to provide climate services to both the public and decision-making audiences through early warning information on drought type, location, extent, and severity. The goal of these diagrams and the web-based tool is to alleviate some management concerns outlined in Hossain et al. (2015) and Sterle et al. (2019), including runoff forecast timing errors, lack of upstream reservoir storage, and reductions in water supply reliability and water quality. These concerns can be start to be addressed through illuminating water supply uncertainties across a range of hydroclimate conditions, enhancing the flexibility of subseasonal-to-seasonal water management practices, and improving coupled atmospheric and hydrologic forecast systems (Siirila-Woodburn et al., 2021). By providing another means to communicate climate information, phase diagrams may help further develop the capacity to identify and to rapidly evaluate underlying vulnerabilities within and between human and natural systems that are susceptible to cascading and compounding effects (Jacobs and Street, 2020; Siirila-Woodburn et al., 2021). The Web-based tool producing the snow drought phase diagrams (https://wrcc-staging.dri.edu/my/climate/snow-drought-tracker) presented herein is concurrently being shared with groups responsible for

communicating snowpack and mountain hydroclimate information to the public such as the U.S. National Weather Service as
well as other water and natural resource managers. Last, snow drought phase diagrams can support innovative ways in thinking
about how weather and hydroclimate variability influence the mountain environment. For example, phase diagrams help us
re-frame snow drought as a time-dependent process rather than a point-in-time concept, helping to contextualize hydrologic
outcomes such as runoff efficiency. They also can be blended with other indicators relevant for ecosystem function (e.g., Con-
tosta et al. (2019)) to explore how snow drought impacts forest health or limnnology as well as changing wildfire behavior
(Alizadeh et al., 2021). Our aim is for this information to aid mountain hydroclimate monitoring and drought early warning
efforts, as well as promote scientific innovation in a changing mountain environment.

*Code availability.* All MATLAB code is available upon reasonable request.

*Author contributions.* **Benjamin J. Hatchett**: Conceptualization, Investigation, Visualization, Software, Formal Analysis, Writing – Origi-
nal Draft, Writing – Review and Editing, Funding Acquisition. **Alan M. Rhoades**: Conceptualization, Investigation, Visualization, Formal
Analysis, and Writing - Review and Editing. **Daniel J. McEvoy**: Conceptualization, Investigation, Funding Acquisition, Writing – Review
and Editing.

*Competing interests.* The authors declare no competing interests.

*Acknowledgements.* We appreciate the constructive reviews and input provided by Adrienne Marshall, an anonymous reviewer, and editor
Veit Blauhut, and feedback on early versions of this manuscript from Michael Anderson, Sasha Gershunov, Tim Bardsley, and Zach Tolby.
Roger Kreidberg provided technical editing and writing advice. Authors Hatchett and McEvoy were supported by the Desert Research In-
stitute Foundation under an Institutional Research Project grant and the National Weather Service. Author McEvoy was supported by the
NOAA Climate Program Office Sectoral Applications Research Program under grant #NA19OAR4310370. Author Rhoades was funded
by the Office of Biological and Environmental Research of the U.S. Department of Energy within the Regional and Global Climate Mod-
eling Program program under the "the Calibrated and Systematic Characterization, Attribution and Detection of Extremes (CASCADE)"
Science Focus Area (award no. DE-AC02-05CH11231) and the "A Framework for Improving Analysis and Modeling of Earth System and
Intersectoral Dynamics at Regional Scales" project (award no. DE-SC0016605).

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
