# Peer review of "Monitoring the Daily Evolution and Extent of Snow Drought"

_Natural Hazards and Earth System Sciences, 2021_

## Author Comment (AC1)

**To:** Natural Hazards and Earth System Sciences
    Reviewer 1
**Re:** Author Responses to Reviews of Manuscript (nhess-2021-193)
**Date:** 12-Nov-2021
* * *
Dear Reviewer 1,
Thank you for your positive feedback and constructive comments. We are addressing and incorporating your suggestions and look forward to submitting a revised manuscript. Our initial responses to your comments are in **bold** with revised text in red.
Sincerely,
Benjamin Hatchett

*Summary* The paper provides a visualization of the progress of snow water equivalent and simultaneously precipitation which allows to determine unset evolution and termination of snow drought this work contributes to monitoring and managing snow droughts the presented approach is an improvement to single date approaches. It also presents an implemented application of the approach in the western US snow monitoring (web tool). I think the approach is very useful and should be disseminated. I really appreciate the idea to not stick to a calendar day to define snow status but rather using the evolution and setting it in perspective to the long-term expectation to be able to define a drought. The paper is written clearly and has a logic structure, however I have some comments that the authors might want to consider before publication.

**We appreciate your positive feedback!**

**I. Main comments**
While I really like the proposed visualization using phase diagrams to simultaneously track snow pack and precipitation, I think the paper does not advertise the approach enough. The opportunities that are opened by using the phase diagrams to visualize could be even more clearly presented. That is in my opinion not because it is not written but because it is hiding a bit in the many "side stories" in the paper.

For instance, now there is relatively much space given to the description of the web-based tool, which I would keep rather short to better present different opportunities these method offers beyond the application it already has in the web tool.

**We appreciate the suggestion to further revise the text on the tool for brevity. Following the note below, we will add some additional discussion about how the tool can inform impacts of snow drought. This is a great suggestion as the goal of the tool is ultimately to support management as well as help researchers understand the temporal evolution of snow drought and its hydrologic impacts.**

On this line, the authors could also clearer discuss, e.g., the impact of the snow evolution on summer low flows; is there a effect in summer at all?

**Good suggestion, we will add additional text on these potential impacts for two of the areas**

**we use as case studies (the Nisqually River watershed and the San Juan Mountains).**

*Comment*

Since not everywhere in the world there is such a snow measuring network like SNOTEL it would be good to put the comparison of station two gridded snow data a bit more in focus which makes this approach more interesting for areas without dense station network.

**Very good point. While the purpose of our paper is not to do comparison with stations versus observations (this has been done by the University of Arizona group who developed the UA SWE product), we appreciate the reviewer's highlighting the need to apply this technique (and the snow drought concept) to poorly instrumented regions. We will add an example of this for the northeastern US using the UA SWE product operating under the assumption that if UA SWE performs reasonably well in the western US it should be at least reasonable for the eastern US. The northeastern US heavily relies on snow for recreation and ecosystem services, but does not have a long-term observational network like the SNOTEL network in the western US. In our revised manuscript, we plan to add a figure showing northeastern US spatial and a phase diagram for a standout snow drought year.**

*Comment*

The authors might consider and discuss for which region regions of the world the presented approach might be a useful approach this is not everywhere the case because winter precipitation is the main driver off summer flow conditions (e.g., Jenicek et al. 2016).

**Thank you for the suggestion. We will add text noting this limitation of our approach as not being ideal for regions or watersheds where precipitation, and not snow, is the dominant driver of summer flow magnitude and variability, such as identified in Jenicek et al. 2016. The northeastern US example above is a good location to highlight this, and we will think of other examples world-wide where this effect may play an important role.**

*Comment*

How sensitive are the seven days longterm that are used to define the drought via quantiles should that be per calendar day or rather day of progress of the snowy season, i.e. day after start of season?

**While developing the percentile windows, we tested three, five, seven, and nine-day windows (with the center date on the day of interest) and did not identify notable sensitivity to this window. As our goal was to remove seasonality (e.g., start of season versus peak snowpack) and still capture day-day variability, we settled on seven days as optimizing the removall of seasonality but also capturing variability by increasing the sample size. In our revision, we will provide a supplementary figure showing the lack of sensitivity.**

**To the reviewer's second point, we will also explore starting the trajectories on the calendar day once there is snow (rather than 1 Nov as we did in the initial submission). This may help avoid large jumps at the start of the season when snowpack can rapidly accumulate.**

**II. Minor Comments**

*Comment*

L21 add that for some areas this very much depending on the climate see for example Jenicek et al. (2016) as opposed to Godsey et al. (2014) where that is true

**Good point, we will add references to the western US and note this limitation in our discussion. New text:**

In the western U.S., spring snowpack is an important predictor of warm season runoff for environmental flows and human consumptive use (Godset et al. 2014; Siirila-Woodburn et al. 2021). However, in lower elevation, humid summer climates, snowpack plays a lesser role in controlling summer hydrologic regimes, i.e., summer flow is less dependent on peak SWE (Jenicek et al. (2016).

*Comment*

L96 regarding the percentiles does this really reflect drought severity as it does for precipitation i.e. are the same range is adequate for snow data?

**Great question. Percentiles are common way to track/identify drought. They normalize the anomaly, meaning it can be applied anywhere and offers a point of reference. Similar percentile-based numbers based on the drought categories identified by the United States Drought Monitor (USDM) have been used by other authors (e.g., Marshall et al. 2020, Siirila-Woodburn et al. 2021, Huning and Aghakouchak 2020), however no precise threshold has been identified for drought severity. Drought means different things to different people or systems. The drought categories identified by the United States Drought Monitor (USDM) is more of a rule of thumb intended to show where likely impacts will occur and based on stakeholder feedback.**

As the purpose of our paper is tracking and monitoring snow drought with the intent of broad applications, we will add text highlighting this potential limitation and the need for future work to further constrain and identify meaningful percentile-based thresholds are by region.

*Comment*

L382/3that a drought onset is approximately at 85% of median snowpack should be made mentioned earlier in the article. **Thank you for the suggestion, we have added text to the introduction addressing this point:**

In addition, no clear definition for snow drought onset exists to our knowledge. For example, Hatchett and McEvoy 2018 used 80% of average SWE and Harpold et al. 2017 used below normal (average) SWE. Depending on the variability of snowpack accumulation, which varies by location and geographic region, snow droughts may occur across a range of percentages of average and are not easily comparable. This motivates the use of a percentile-based approach to facilitate regional comparisons.

*Comment*

Figure 2 left panel: I would suggest to put the annotations outside the plot and just add a line pointing at the labeled parts particularly under the left side of the plot it is very dense and is better readable outside the plot; right panel: Some of the text is hard to read consider larger font or darker color

**Thank you for the suggestions. We have cleaned up the plot as suggested and agree it has improved. We are also adjusting the phase diagram plots following the second reviewer's comments and will include the new plots in our revised manuscript.**

**III. Technical comments by line (L) number**
L9 add with before streamflow

Added "with"

L22 "in efforts of 89% of areas western United States" please reword not clear

Good suggestion, we apologize for the unclear statement. We have revised as follows:

"Snow water equivalent loss is projected to decline by 40-60% in the western United States (wUS) by end-century (Woodburn et al. 2021). For regions historically characterized by a seasonal snowpack, these declines are projected to reduce drought prediction skill (Livneh and Badger 2020)."

L23 change "will" to "are predicted to"

Change made, thank you.

L30 defined or not? maybe rather use roughly or about

Correct, "defined as". To make this more clear, we revised to: "defined as November–May".

L41 to be for varied user groups

Good catch, revised to: "...conditions **to** varied user groups..."

L44 allows to allow; "track their" referring to signals? Not clear

**Thanks for noting this should be made more clear. We revised to:**

"allow a user to track snowpack and precipitation evolution through the cool season or the entire water year (WY)."

L41-44 consider breaking the sentence into for better readability

**Good suggestion. We broke this into two separate sentences:**

"These challenges, and the need to communicate mountain hydroclimate conditions to varied user groups (e.g., the National Weather Service, natural resource managers and other decision makers Marshall et al. 2020), illustrate the need for easily-accessible, informative data visualization approaches. Ideally, these visualizations capture the signals of interest for decision-relevant contexts and allow a user to track snowpack and precipitation evolution through the cool season or the entire water year (WY)."

L46 remove also before demonstrate

**Change made, thank you.**

L58 add reference to Table 1

**Table reference was added, thank you.**

L64 What was the motivation to use this station? Why not one for each snow climate?

**We selected this pair of SNOTEL and streamgage stations to use this region as proof of concept for sensitivity of runoff to varied snow drought conditions. We revised the text to point this out more clearly:**

Last, we highlight an example of how snow drought conditions influence streamflow and to show how to link phase diagrams with hydrologic outcomes. To do this, we acquired daily streamflow for WYs 1943–2019 from the U.S. Geological Survey Gage 12082500, located on the unimpaired Nisqually River, near the Paradise, Washington SNOTEL site (Fig. 1b).

Figure 1: is it possible to push elevation legend bar down a bit?

**Elevation bar has been moved down.**

L68 add space before "Phase "

**Space added.**

L69 is → are

**Change made, thank you.**

L77 are → is

**Change made, thank you.**

L104 remove the before 31

**This has been removed, thank you.**

L 427 National Weather Service specify that this is for the US

**We have clarified this, thank you.**

Added References:

Godsey, S. E., Kirchner, J. W., and Tague, C. L.: Effects of changes in winter snowpacks on summer low flows: case studies in the Sierra Nevada, California, USA, Hydrol. Process., 28, 5048–5064,doi:10.1002/hyp.9943, 2014.

Jenicek, M., Seibert, J., Zappa, M., Staudinger, M., and Jonas, T.: Importance of maximum snow accumulation for summer low flows in humid catchments, Hydrol. Earth Syst. Sci., 20, 859–874, https://doi.org/10.5194/hess-20-859-2016, 2016.

---

## Author Response (AR2)

**To:** Natural Hazards and Earth System Sciences
    Reviewer 1
**Re:** Author Responses to Reviews of Manuscript (nhess-2021-193)
**Date:** 27-Dec-2021
* * *
Dear Reviewer 1,
Thank you for your positive feedback and constructive comments. We have addressed and incorporated your suggestions into our revised manuscript. Our responses to your comments are in **bold** with revised text in red.
Sincerely,
Benjamin Hatchett

*Summary*

The paper provides a visualization of the progress of snow water equivalent and simultaneously precipitation which allows to determine unset evolution and termination of snow drought this work contributes to monitoring and managing snow droughts the presented approach is an improvement to single date approaches. It also presents an implemented application of the approach in the western US snow monitoring (web tool). I think the approach is very useful and should be disseminated. I really appreciate the idea to not stick to a calendar day to define snow status but rather using the evolution and setting it in perspective to the long-term expectation to be able to define a drought. The paper is written clearly and has a logic structure, however I have some comments that the authors might want to consider before publication.

**We appreciate your positive feedback!**

**I. Main comments**
While I really like the proposed visualization using phase diagrams to simultaneously track snow pack and precipitation, I think the paper does not advertise the approach enough. The opportunities that are opened by using the phase diagrams to visualize could be even more clearly presented. That is in my opinion not because it is not written but because it is hiding a bit in the many "side stories" in the paper.

**We see the reviewer's point regarding the side stories, however we are unsure as how to better advertise the approach. The numerous examples were intended to demonstrate the various ways this visualization approach could inform drought monitoring and early warning (as well as help link to snowpack-related impacts, i.e., the Pacific Northwest snow drought and the Utah Avalanche example). We are completely open to specific suggestions on how to better advertise our approach in lieu of examples with side-stories, but we believe our present text provides a reasonable range of illustrations of how this technique could be applied to real-world problems.**

For instance, now there is relatively much space given to the description of the web-based tool, which I would keep rather short to better present different opportunities these method offers beyond the application it already has in the web tool.

We appreciate the suggestion to further revise the text on the tool, which we have done. Following the note below, we added additional discussion about how the tool can inform impacts of snow drought. This is a great suggestion as the goal of the tool is ultimately to support management as well as help researchers understand the temporal evolution of snow drought and its hydrologic impacts. We revised our limitations section to better highlight the need for locally-relevant drought thresholds (see text pertaining to your second Minor Comment below).

On this line, the authors could also clearer discuss, e.g., the impact of the snow evolution on summer low flows; is there a effect in summer at all?

**Good suggestion-thank you-we added additional text on these potential impacts, please see our more detailed response to Minor Comment 1.**

*Comment*

Since not everywhere in the world there is such a snow measuring network like SNOTEL it would be good to put the comparison of station two gridded snow data a bit more in focus which makes this approach more interesting for areas without dense station network.

**The reviewer raises an important point. While the purpose of our paper is not to do comparison with stations versus observations (this has been done by the University of Arizona group who developed the UA SWE product, please see their work (Zhang et al. 2018 and references therein, e.g., Broxton et al. 2016)), we appreciate the reviewer's highlighting the need to apply this technique (and the snow drought concept) to poorly instrumented regions.**

**In addition to the comparison of gridded SWE at the basin scale to SNOTEL stations in the manuscript (section 4.5.1), we added an additional example of this for the northeastern US using the UA SWE product (new section 4.5.2). The northeastern US heavily relies on snow for recreation and ecosystem services, but does not have a long-term observational network like the SNOTEL network in the western US. Our new figure and text are below and show four examples for the Susquehanna Basin:**

**Section 4.7.1 Basin-Averaged Phase Diagrams for an Observation-Limited Region** The Susquehanna River Basin of the northeastern U.S. includes both seasonal and ephemeral snowpacks (Hatchett, 2021); Figure 1a) that occur during extended winter (December-March; Figure 1b). Although snowpack in the Susquehanna Basin is not as critical for downstream water availability as in the western U.S., it supports winter recreation and is a critical component for ecosystems function in northern forests (Contosta et al., 2019). The Susquehanna is highly susceptible to flooding, especially during snowmelt or rain-on-snow events that contribute to enhanced runoff (Yarnal et al., 1997). Even in locations without long-term snowpack monitoring data, basin-averaged phase diagrams can be applied to highlight a range of drought outcomes. WY1988 demonstrates a dry winter ($< 25^{th}$ percentile precipitation) that oscillated between moderate (D1) snow drought and "Dry but Snowy" (up to $67^{th}$ percentile SWE) conditions (Figure 1c). While WY1994 started off the winter season in moderate (D1) warm snow drought conditions, substantial improvement occurred from mid-December onward with consistently cold conditions and a wetter-than-average February (Figure 1d) bringing a near-record snowpack late

[Figure]

Figure 1: **(a)** Peak snow water equivalent (SWE) in the Susquehanna River Basin **(b)**. Daily SWE climatology and time series of four water years. Phase diagrams for **(c)** Water Year (WY) 1988, **(d)** WY 1994, **(e)** WY 1995, and **(f)** WY 2016.

spring snowpack (Figure 1b). The snowpack and saturated soils developed during the "Big Year" of WY1994 culminated in a large flood event in early April (Marosi and Pryor 2000). A similar situation occurred in WY1996 (not shown) with extremely wet and snowy conditions leading up to a warm, windy, and wet January storm that coincided with saturated as well as frozen soils that contributed to overland flow and widespread flooding (Yarnal et al. 1997). WY1995

demonstrates another dry snow drought outcome, though with more severe snow drought conditions (as low as exceptional (D4)) but a slightly wetter overall winter than WY1988 (Figure 1e). In 2016, a notable warm snow drought year occurred in the Northeastern U.S. with record temperatures (Sweet et al. 2017) and closer-to-average winter precipitation leading to dry and warm snow drought (trajectories plotting to the right of the 1:1 line; Figure 1f). The exceptionally low snowfall (much of the winter ranged from D1 to D4) exacerbated drought conditions during the following drier-than-normal warm season, leading to 30%-70% crop losses (Sweet et al. 2017). The Susquehanna examples further indicate gridded products can be applied to create meaningful phase diagrams at the basin scale to track snow drought through time.

*Comment*

The authors might consider and discuss for which region regions of the world the presented approach might be a useful approach this is not everywhere the case because winter precipitation is the main driver off summer flow conditions (e.g., Jenicek et al. 2016).

**Thank you for the suggestion. We added text noting this limitation of our approach as not being ideal for regions or watersheds where precipitation, and not snow, is the dominant driver of summer flow magnitude and variability, such as identified in Jenicek et al. 2016. The northeastern US example above is a good location to highlight this. We also noted areas in the Colorado River Basin where summer convective precipitation has been shown to augment streamflow or ET (Carroll et al., 2020).**

*Comment*

How sensitive are the seven days longterm that are used to define the drought via quantiles should that be per calendar day or rather day of progress of the snowy season, i.e. day after start of season?

**The percentiles were calculated for each calendar day within the snow season (1 October-31 May) where the start of the snowy season is defined as 1 October (see text in Methods). Because SWE is a state variable, each calendar day it is calculated for represents the cumulative outcome of the prior days within the season (implying the calendar day and day of progress following the start of the season are the same). While developing the percentile windows, we tested three, five, seven, and nine-day windows (with the center date on the day of interest) and did not identify notable sensitivity to this window. As our goal was to reduce seasonality (e.g., start of season versus peak snowpack) and still capture day-day variability, we settled on seven days as optimizing the removal of seasonality but also capturing variability by increasing the sample size. We also added a brief note to the methods about this approach to reduce seasonality effects:**

*"For each station, we calculated daily percentiles of accumulated precipitation and SWE from 1 November to 30 April or 31 May using a seven-day moving window centered on each calendar day to reduce seasonality effects (Montecinos et al., 2017, Shortridge et al., 2019."*

We added citations for the moving window approach to removing seasonality: Montecinos, A., Muñoz, R. C., Oviedo, S., Martínez, A., and Villagrán, V.: Climatological Characterization of Puelche Winds down the West- ern Slope of the Extratropical Andes Mountains Using the NCEP

Climate Forecast System Reanalysis, Journal of Applied Meteorology and Climatology, 56, 677 – 696, https://doi.org/10.1175/JAMC-D-16-0289.1, 2017.

Shortridge, A., Hatchett, B. J., and Gustin, M. S.: Rise in coincidence of extreme heat in Nevada's largest urban areas, Journal of Nevada660 Water Resources Association, Winter, 5–28, https://doi.org/https://doi.org/10.22542/jnwra/2019/1/1, 2019.

To show this, we performed analysis for two locations (Mt Rose Ski Area, Nevada and Red Mountain, Colorado) and four dates, showing both accumulated precipitation-to-date and snow water equivalent. The results show that there is little sensitivity to the window of time used (an of course one could argue this is not a necessary step based on these results) for small windows (up to two weeks). However, note the need to calculate percentiles based on the date of interest rather than over the course of the majority of the snow season (181-day window; shown by purple lines), as doing so introduces high or low biases to both accumulated precipitation and snow water equivalent at all points in the year. We appreciate the reviewer's suggestion to explore this more deeply.

[Figure]

Figure 2: Range of snow water equivalent values (y-axis) as a function of percentiles (x-axis) at Mt. Rose Ski Area, Nevada (NV), for varying calculation windows: **(upper left)** 1 December, **(upper right)** 1 February, **(lower left)** 1 March, and **(lower right)** 1 April.

[Figure]

Figure 3: Range of snow water equivalent values (y-axis) as a function of percentiles (x-axis) at Red Mountain, Colorado (CO), for varying calculation windows: **(upper left)** 1 December, **(upper right)** 1 February, **(lower left)** 1 March, and **(lower right)** 1 April.

[Figure]

Figure 4: Range of precipitation values (y-axis) as a function of percentiles (x-axis) at Mt. Rose Ski Area, Nevada (NV), for varying calculation windows: **(upper left)** 1 December, **(upper right)** 1 February, **(lower left)** 1 March, and **(lower right)** 1 April.

[Figure]

Figure 5: Range of precipitation values (y-axis) as a function of percentiles (x-axis) at Red Mountain, Colorado (CO) for varying calculation windows: **(upper left)** 1 December, **(upper right)** 1 February, **(lower left)** 1 March, and **(lower right)** 1 April.

**To the reviewer's second point, we decided to shift the starting point of the the trajectories to 1 Nov instead of 1 Oct for the Sierra Nevada locations to account for the often dry falls (where 70% of the time it is dry, which would give the wrong impression of above normal precipitation). This seems to help avoid large jumps at the start of the season when snowpack/precipitation can rapidly accumulate. We also ended several trajectories earlier (30 April rather than 31 May) for the same reason. In both early and late season, the percentiles, like a percent of average approach, are not as meaningful when the distribution tends towards zero. For instance, if 70% of the time there is no SWE during May, but April ended at 30% SWE (in snow drought as we define it), the trajectory would evolve towards a 'better' outcome as the zeros become increasingly frequent and drive the percentiles higher.**

**II. Minor Comments**

*Comment*

L21 add that for some areas this very much depending on the climate see for example Jenicek et al. (2016) as opposed to Godsey et al. (2014) where that is true

**Good point, we added this limitation to our discussion. New text:**

In addition to the development of locally-relevant thresholds, the identification of watershed-level sensitivity of streamflow to snow drought type and magnitude can add additional insight to drought monitoring, early warning, and how continued warming will influence mountain hydrology. Many western U.S. watersheds are characterized by dry summer conditions, relying upon snowmelt-derived groundwater recharge to maintain warm season streamflow (e.g., Godsey et al., 2013). However, exceptions occur in humid summer climates where snowpack plays a lesser role in controlling summer hydrologic regimes, i.e., summer flow is less dependent on peak SWE (Jenicek et al., 2016). This may be the case in lower elevation watersheds in the Pacific Northwest or in regions impacted by the North American Monsoon, a warm-season precipitation regime that Carroll et al. (2020) found augments low elevation evapotranspiration and high elevation streamflow. Watershed-specific hydroclimatic analyses could help identify and rank the risks snow drought poses to water availability and could help inform adaptation strategies to offset reductions in water availability from projected snowpack decline (Siirila-Woodburn et al., 2021).

Added citations:

Carroll, R. W. H., Gochis, D., and Williams, K. H.: Efficiency of the Summer Monsoon in Generating Streamflow Within a Snow-Dominated Headwater Basin of the Colorado River, Geophysical Research Letters, 47, e2020GL090856, https://doi.org/https://doi.org/10.1029/2020GL090856, 2020

Godsey, S. E., Kirchner, J. W., and Tague, C. L.: Effects of changes in winter snowpacks on summer low flows: case studies in the Sierra Nevada, California, USA, Hydrol. Process., 28, 5048–5064,doi:10.1002/hyp.9943, 2014.

Jenicek, M., Seibert, J., Zappa, M., Staudinger, M., and Jonas, T.: Importance of maximum snow accumulation for summer low flows in humid catchments, Hydrol. Earth Syst. Sci., 20, 859–874, https://doi.org/10.5194/hess-20-859-2016, 2016.

*Comment*

L96 regarding the percentiles does this really reflect drought severity as it does for precipitation i.e. are the same range is adequate for snow data?

**Great question, and a difficult one. Percentiles are common way to track/identify drought. They normalize the anomaly, meaning it can be applied anywhere and offers a point of reference. Similar percentile-based numbers based on the drought categories identified by the United States Drought Monitor (USDM) have been used by other authors (e.g., Marshall et al. 2020, Siirila-Woodburn et al. 2021, Huning and Aghakouchak 2020), however no precise threshold has been identified for drought severity. Drought means different things to different people or systems. The drought categories identified by the United States Drought Monitor (USDM) is more of a rule of thumb intended to show where likely impacts will occur and based on stakeholder feedback.**

As the purpose of our paper is tracking and monitoring snow drought with the intent of broad applications, we included text highlighting this potential limitation and the need for future work to further constrain and identify meaningful percentile-based thresholds are by region (*italics for emphasis*:

 Collaborations with natural resource managers, practitioners, and decision makers will be instrumental in the *development of locally- or regionally-specific snow drought thresholds*.

**Elsewhere, we made sure to point out the benefit of percentiles over percent of median/mean by allowing comparisons between locations:** On the other hand, percentiles provide perspective for less-familiar users to understand the state of the snowpack. Percentiles also allow comparisons between locations in terms of snow drought severity.

*Comment*

L382/3 that a drought onset is approximately at 85% of median snowpack should be made mentioned earlier in the article.

**Thank you for the suggestion, we have added text to the introduction addressing this point:**

 In addition, no clear definition for snow drought onset exists to our knowledge. For example, Hatchett and McEvoy 2018 used 80% of average SWE and Harpold et al. 2017 used below normal (average) SWE. Depending on the variability of snowpack accumulation, which varies by location and geographic region, snow droughts may occur across a range of percentages of average and are not easily comparable. This motivates the use of a percentile-based approach to facilitate regional comparisons.

*Comment*

Figure 2 left panel: I would suggest to put the annotations outside the plot and just add a line pointing at the labeled parts particularly under the left side of the plot it is very dense and is better readable outside the plot; right panel: Some of the text is hard to read consider larger font or darker color

**Thank you for the suggestions. We have cleaned up the plot as suggested (we used a number instead of another arrow pointing to the text, and titled the list 'Synthesis of WY 2020') and**

**believe it has improved. We also adjusted the phase diagram plots to include a 1:1 line in the third quadrant (dry snow drought quadrant) following the second reviewer's comments. We also increased the font size in the right hand panel. The revised figure is below:**

[Figure]

Figure 6: **(a)** Annotated phase diagram showing 1 October 2019 to 31 May 2020 at the Central Sierra Snow Laboratory (CSS Lab), California. **(b)** Conceptual phase diagram showing potential physical interpretations of seasonal evolution of various trajectories.

**III. Technical comments by line (L) number**

L9 add with before streamflow

Added "with"

L22 "in efforts of 89% of areas western United States" please reword not clear

Good suggestion, we apologize for the unclear statement. We have revised as follows:

"Snow water equivalent loss is projected to decline by 40-60% in the western United States (wUS) by end-century (Siirila-Woodburn et al. 2021). For regions historically characterized by a seasonal snowpack, these declines are projected to reduce drought prediction skill (Livneh and Badger 2020)."

L23 change "will" to "are predicted to"

Change made, thank you.

L30 defined or not? maybe rather use roughly or about

Correct, "defined as". To make this more clear, we revised to: "defined as November–May".

L41 to be for varied user groups

Good catch, revised to: "...conditions **to** varied user groups..."

L44 allows to allow; "track their" referring to signals? Not clear

**Thanks for noting this should be made more clear. We revised to:**

"allow a user to track snowpack and precipitation evolution through the cool season or the entire water year (WY)."

L41-44 consider breaking the sentence into for better readability

**Good suggestion. We broke this into two separate sentences:**

"These challenges, and the need to communicate mountain hydroclimate conditions to varied user groups (e.g., the National Weather Service, natural resource managers and other decision makers Marshall et al. 2020), illustrate the need for easily-accessible, informative data visualization approaches. Ideally, these visualizations capture the signals of interest for decision-relevant contexts and allow a user to track snowpack and precipitation evolution through the cool season or the entire water year (WY)."

L46 remove also before demonstrate

**Change made, thank you.**

L58 add reference to Table 1

**Table reference was added, thank you.**

L64 What was the motivation to use this station? Why not one for each snow climate?

**We selected this pair of SNOTEL and streamgage stations to use this region as proof of concept for sensitivity of runoff to varied snow drought conditions. We revised the text to point this out more clearly:**

Last, we highlight an example of how snow drought conditions influence streamflow and to show how to link phase diagrams with hydrologic outcomes. To do this, we acquired daily streamflow for WYs 1943–2019 from the U.S. Geological Survey Gage 12082500, located on the unimpaired Nisqually River, near the Paradise, Washington SNOTEL site (Fig. 1b).

Figure 1: is it possible to push elevation legend bar down a bit?

**Certainly. The elevation bar has been moved down.**

L68 add space before "Phase "

**Space added.**

L69 is → are

**Change made, thank you.**

L77 are → is

**Change made, thank you.**

L104 remove the before 31

**This has been removed, thank you.**

L 427 National Weather Service specify that this is for the US

**We have clarified this, thank you.**
* * *
Dear Dr. Marshall,

Thank you for your positive feedback and constructive comments. To the best of our ability, we have addressed and incorporated your suggestions in our revised manuscript submission. Our responses to your comments are in **bold** with revised text in red.

Sincerely,

Benjamin Hatchett

**I. General comments**

This manuscript introduces a novel visualization method for illustrating snow drought evolution using phase diagrams. The authors illustrate the use of this method through a series of case studies, and evaluate the utility of the method for in situ snow observation data as well as gridded data products. They also document a web-based tool to allow users to view these snow drought phase diagrams for locations and time periods of interest. The manuscript is well-written and appropriate for NHESS, and I like the idea of using phase diagrams for snow drought identification. I have quite a few specific comments, but none of them require fundamental changes to the manuscript. The most important specific comments below are to:

*Comment*

Consider adding a 1:1 line to the phase diagrams and additional or alternative designations for warm snow drought

**We like this idea and have added a 1:1 line to the revised figures. We added a brief discussion about the idea of alternative designations for snow droughts with below median precipitation regimes that also indicate a 'flavor' of warm snow drought. New text added to Section 4.2:**

An additional case of 'dry-but-warm' snow drought can also occur when trajectories are in the dry quadrant but sit on the wetter side of a 1:1 line (dashed line in the dry quadrants). These conditions can be indicative of drier-than-average conditions overall with warmer-than-average precipitation events limiting snowpack accumulation.

*Comment*

Reconsider the implications of data presented in Figure 6 (on basin-averaged gridded data vs point scale in situ data)

**We agree with this point. Please see our responses later in the specific comments.**

**II. Specific Comments by Line (L) number**

*Comment*

Line 19 – I think the idea that a snow-to-rain transition leads to less runoff efficiency is somewhat contested; I might also consider referencing Barnhart et al. (2020) and nuancing this statement a bit.

Barnhart, Theodore B., Christina L. Tague, and Noah P. Molotch. "The Counteracting Effects of Snowmelt Rate and Timing on Runoff." Water Resources Research 56, no. 8 (2020): e2019WR026634. https://doi.org/10.1029/2019WR026634.

**Thank you for the reference and the suggestion to alter our text to address this issue. We revised the text to highlight the alternative explanations of how a transition from snow to rain will alter mountain hydrology with respect to runoff processes and included your suggested citation (thanks!). Our revised (and added; in bold) text is as follows:**

As rain falls instead of snow, runoff could become less efficient (Berghuijs et al., 2014) as water is no longer stored in the snowpack and as warming increases atmospheric water demand (Fisher et al., 2017). **However, seasonal shifts in energy availability altering plant available water storage (Barnhart et al., 2020) may counteract reduced**

**snowmelt rates and changes in rain and snow partitioning, potentially buffering runoff changes in energy-limited (colder) environments.**

*Comment*

Line 21 – maybe specify "in snow-dominated regions".

**Change made, thank you.**

*Comment*

Table 1 – I don't think Table 1 is referenced in the text;

**Thank you for pointing out our omission to the table citation; this has now been fixed in the text.**

*Comment*

could you specify the source of the "snow climate" designation? I don't disagree with any of these designations, but think it might be helpful for readers. Or, describe what you mean by each when you reference these climates in the text (Line 59).

**Yes, and thank you for the suggestion. We added a citation to the source of these classifications (Mock and Birkeland (2000) as our designations pertain largely to continentality (distance from the ocean).**

**Added citation:**

Mock, C. J., and Birkeland, K. W. (2000). Snow Avalanche Climatology of the Western United States Mountain Ranges, Bulletin of the American Meteorological Society, 81(10), 2367-2392. Retrieved Nov 12, 2021, from https://journals.ametsoc.or 0477_2000_081_2367_sacotw_2_3_co_2.xml

*Comment*

Line 121 – So the "dry but snowy" conditions could essentially either indicate anomalously cold storm events for the season, or a seasonal shift in precipitation timing (towards colder), right? Consider whether this might be a helpful rephrasing of what you already have here.

**Great suggestion to add additional nuance to the discussion here. We've added the following text:**

"In addition, a shift in precipitation timing into the colder months of the season could also drive a leftward shift towards the "dry but snowy regime" (Gershunov et al., 2019)."

**Added citation:**

Gershunov, A., Shulgina, T., Clemesha, R.E.S. et al. Precipitation regime change in Western North America: The role of Atmospheric Rivers. Sci Rep 9, 9944 (2019). https://doi.org/10.1038/s41598-019-46169-w

*Comment*

Figure 2a – Do you think it would be appropriate to add a 1:1 line? I found myself wanting to know for a given precipitation percentile if the snowpack for that date was above or below the snowpack percentile. I know that warm snow droughts are typically defined as having approximately normal precip with below-normal SWE, but I also wonder about identifying cases where SWE is proportionally lower than precipitation – e.g., if precipitation was 30% of normal, and SWE was 5% of normal, wouldn't we still want to think of this as a warm snow drought at least to some extent? Maybe it needs a different designation? This would fall below a 1:1 line, but not in the lower right quadrant.

**We like this suggestion and have added a 1:1 line to the revised figures. It makes lots of sense for the lower left quadrant to help differentiate dry-but-warm snow droughts from dry-but-cold snow droughts. This may lead to sub-categories that could have interesting hydrological or recreational implications. An example: both WY2015 and WY2020 (and WY2021) were dry, WY2015 was on the right of the 1:1 line (dry-but-warm) whereas the other years were dry-but-cold. From personal experience, the ski conditions and other winter recreation conditions in dry-but-cold scenarios (2020, 2021 in Tahoe) were far superior to dry-but-warm snow droughts (2015).**

*Comment*

Line 137 - "seasonally induced shifts in solar insolation" It makes sense to me that this causes melt, but why would it cause a decrease in the percentile of SWE, given that these solar insolation shifts happen at roughly the same time every year (unless there were anomalous clear skies)?

**The idea we intended to explain is that normally this is still the accumulation season and apologize for not being clear in our explanation. We interpret the dry conditions and anomalously clear skies to increase solar radiation, allowing the snow-albedo feedback (which is presumably more effective at melting/ripening snow with the increasing sun angle and duration of daylight at this time of year) to accelerate declines in snow percentile. Climatologically, SWE should be accumulating and even constant SWE this time of year would result in declining SWE percentiles with time. This is likely too much nuance, so ultimately we removed the text about solar radiation to avoid confusion.**

*Comment*

Line 199 – Suggest specifying that "comparable snow drought conditions" refers to SWE percentiles only.

**Good point, we added:**

"(with respect to snowpack percentiles)"

*Comment*

Line 201 – I have not seen the Native name for Ti'Swaq' used before in scientific/snow hydrology literature and I appreciate it! Might this be appropriate for any other place names you use?

**Definitely, and we altered other names in the text for locations we can find Indigenous place names, such as Da Ow (Lake Tahoe).**

*Comment*

Line 226 – Yes, although depending on how you define hydroclimate, Marshall et al. (2019) showed projected decreases in interannual variability of peak SWE in many locations.

**Thank you for pointing out the need to be more explicit in how we define hydroclimate (our intent was more focused on precipitation here) and to add the point about decreasing peak SWE variability. We have revised the text as follows:**

"Enhanced interannual precipitation variability and decreased peak SWE variability are expectations of a warming climate (Boer et al. 2009, Pendergrass et al. 2017, Marshall et al. 2019)."

*Comment*

Figure 6 – It seems like the two basin-averaged snow drought phase diagrams are much more similar to each other than either is to its respective SNOTEL site. It seems the text interprets this figure as saying that basin-averaged SWE diagrams are useful when no in situ observation exists; while I don't disagree that this may be the case, I'd be more inclined to comment on either the limitations of SNOTEL sites for representing basin characteristics or the impacts of spatial averaging on how we interpret these data.

**Good suggestions, and thank you for bringing this point up. While our initial interpretation intended to demonstrate the basin-average SWE diagrams will work in lieu of in-situ observations (i.e., they appear reasonable), the two components of your comment are important to include in our discussion. From my understanding, SNOTEL stations are placed with the intention to be representative of the snow-dominated portion of the watershed, but they are limited by site access as well as other constraints such as wilderness designations, and thus may not be fully representative of the basin characteristics.**

**To your second point, the spatial averaging could have a notable impact on the behavior of the phase diagram owing to basin hypsometry, accumulation patterns (and whether these are well-represented by the spatially-distributed snowpack model), and other factors such as land surface and vegetation characteristics.**

**We revised this section of manuscript to include this discussion, as this may help users better interpret differences between phase diagrams created using in-situ observations, gridded products, or other spatially-distributed blended observational products utilizing remotely sensed data. Our revised text is as follows:**

By the end of the cool season (30 April), both regions showed similar SWE percentiles (within 10 points). Precipitation percentiles were different by only a few points in the Tuolumne but by over 20 points in the Upper Yuba. These differences, as well as the differences in trajectories throughout the winter, likely reflect the signal of orographically enhanced precipitation-or rain shadowing in the case of Virginia Lakes, which lies on the leeside of the Sierra Nevada crest. The CSS Lab was substantially wetter compared to the Upper Yuba during March and April of WY2020, whereas Virginia Lakes showed a similar trajectory but slightly shifted towards drier percentiles. The SNOTEL stations, likely by virtue of their location compared to the watershed hypsometry, may report higher SWE percentiles compared to the basin average if the SNOTEL station is located at a high elevation (CSS Lab) or lower SWE percentiles if the station is found at a middle elevation on the leeward (dry) side (Virginia Lakes). These effects could be exacerbated at the basin scale by the inclusion of lower elevation terrain whose snow rapidly melts out during drought years (potentially accelerated by snow-albedo feedbacks, Groisman et al. 1994). While SNOTEL stations are placed with the intention of being maximally representative for water resources given siting restrictions, the limitations created by varying topography and snowpack accumulation patterns warrants care in extrapolating data from a station to a basin or averaging across a basin in the absence of station data. Despite these differences stemming from the challenges of using limited in-situ observations, the basin-aggregated phase diagrams appear reasonably representative in capturing the broader hydroclimate conditions interpreted from phase trajectories.

*Comment*

Line 276, 279 – You refer to making the phase diagrams "meaningful" in complex terrain – could this get a little more specific and denote "meaningful to what?"

Clarify this: Substitute "extract maximum information value for application of interest" What scale or process are we interested in?

**Yes, we appreciate the suggestion to be more specific as well as the suggested revision. Our intent was to get at meaningful scales or aspects of the watershed for management or science, i.e., do we need to examine the basin as a whole, or focus on a particular elevation band or subwatershed?**

**Using the suggested text, we added details on what we mean by meaningful. New text as follows:**

"The challenge is how to aggregate spatial information to extract maximum information value for the application of interest (e.g., water management, avalanche forecasting, ecosystem processes) regarding the state and evolution of snowpack conditions at relevant scales (e.g., the full watershed, a sub-basin, or within specific elevation bands) in complex terrain."

**Following this line of thought, we also revised the second use of "meaningful" later in the paragraph: "Creating meaningful phase diagrams** *for varied management and scientific applications* **using spatially distributed information is the primary goal of our ongoing research.**

*Comment*

Figure 8 – I like this figure for giving a time series version, but it feels a little disappointing at this point in the manuscript to resimplify to percentile-based snow droughts. Could this figure be expanded (either with additional panels or additional colors, as in Figure 7) to differentiate between warm and dry snow droughts (as you suggest in Line 323)?

**You bring up a good point about the awkwardness of re-simplifying to percentile-based snow droughts at this location in the manuscript. Instead of making the figure more complex by expanding it (we do appreciate the suggestion!) given that the length of the manuscript is becoming unwieldy (¿8,000 words) and that this figure is not central to the phase diagram concept (also making for an even longer gap in the text between the phase diagrams and the web tool),** we opted to remove this figure and its associated text. **We think it would be better utilized in a separate manuscript where we can expand discussion of the information shown in sufficient detail to do it justice.**

*Comment*

Line 320 – I suggest using relative p-values (e.g., $p < 0.001$), rather than giving a precise number; it risks placing more importance on the numerical p-value than is warranted. It's interesting that the April trend is smaller but more consistent (significant) than the December trend; you could point out that this is probably because April trends integrate warm temperatures over the course of the water year, while December trends are more impacted by stochastic precipitation and warm/cold events.

**Your point about placing too much importance on the value is well-taken. We revised to report relative p-values in the figure, however please note (and see above comment) that this figure was removed.**

**We added a line of text following your thoughts about the April vs. December trends:**

The smaller but more significant April trend compared to December may reflect the impact of integrating warming temperatures throughout the season.

*Comment*

Line 370 – Percent of average also gets weird at the end of the season, either in big snow years (percent of average can be huge or infinite, if average is zero), or in years with earlier than average melt date (percent of average can be very small for a few days, but this may exaggerate the apparent low snow conditions).

**Yes, and seeing some of the huge numbers reported in the media or elsewhere was part of the motivating factor to move away from percent of average.**

*Comment*

Line 400 – I appreciated this discussion of the differences between percent of medians and percentiles.

**Thank you! We found this example to be eye-opening with respect to how different these values can be for different locations.**

*Comment*

Line 416 – You mention climate change here; it may also be appropriate to provide a brief mention somewhere of the impact of the period of record on these percentiles. Should the period of record used to calculate percentiles be allowed to extend over non-stationary climates? When comparing multiple sites, how should users account for potential differences in the period of record used for percentile calculations?

**These are important and relevant points that introduce challenges to our analysis (multiple site issues) especially as the climate changes (non-stationarity issue). We will address both of these in the revised manuscript, but here are our initial responses. While comprehensively handling these issues (i.e., solving them) is beyond the scope of our work, bringing up these issues are important for highlighting the limitations of our approach.**

**To the first point, on non-stationarity, it seems salient to use a fixed historic period based on water management assumptions from which to calculate percentiles. This is difficult in practice given that water rights allocations and water management frameworks were developed between the late 1800s and mid-1900s (culminating in the mid-20$^{th}$ century-era of dam building) given the only snowpack information for this period includes the monthly snow courses. Other work by one of the authors (Hatchett et al. 2015, 2018) in the western Great Basin (eastern Sierra Nevada) suggests that the period 1971-2000 is on par with the wettest period in the last 3,900 years. In addition, for the same region, Sterle et al. (2019) highlighted the period spanning 2010-2017 included years that were on par with the average wettest and driest climates at millennial timescales. In the Upper Colorado Basin, despite persistent drought in recent decades, several extremely wet and snowy years have been observed (e.g., 1993, 2011, 2019). It thus seems reasonable to include data for these years to capture the range of natural variability, however observed snowpack decline (e.g., Mote et al. 2018; Siirila-Woodburn et al. 2021) is likely shifting snowpack distributions towards lower values. We are thus left with a difficult choice as to when should additional years in the percentile calculation no longer be added to ensure we don't appear to be out of snow drought when in reality, to paraphrase the late Kelly Redmond, "not enough water is available to meet needs (human or ecosystem)". The reviewer has brought up a difficult but critical question that we will grapple with as we revise the manuscript.**

Hatchett, B. J., Boyle, D. P., Putnam, A. E., and Bassett, S. D. (2015), Placing the 2012–2015 California-Nevada drought into a paleoclimatic context: Insights from Walker Lake, California-Nevada, USA, Geophys. Res. Lett., 42, 8632–8640, doi:10.1002/2015GL065841

Hatchett, B.J., Boyle, D.P., Garner, C.B., Kaplan, M.L., Bassett, S.D., and Putnam, A.E. (2021), Sensitivity of a western Great Basin terminal lake to winter northeast Pacific storm track activity and moisture transport, From Saline to Freshwater: The Diversity of Western Lakes in Space and Time, Scott W. Starratt, Michael R. Rosen. Geological Society of America Special Papers 536, https://doi.org/10.1130/2018.2536(05)

**Following your suggestion for a brief mention, we added a note to address this comment (normal font for context):** "They can also be applied to investigate how climate change may alter phase diagram trajectories and/or residence times of WY snowpack conditions in particular quadrants of the phase diagram. *However, as warming shifts the distribution of early, late, and spring peak snowpack towards lower values, it is worth considering holding the period from which percentiles are calculated constant over a management-relevant time period. This would reflect*

*the historic conditions from which water management made assumptions about snowpack and water availability (Siirila-Woodburn et al. 2021).*"

**Regarding the differing periods of records for stations, we will add text to the phase diagram figures in the web-tool (here readers can refer to Table 1 with periods of records) such that readers are made aware of the period of record used. How users account for these differences is a perennial challenge in applied/service climatology, as knowing long-term context allows users to differentiate between a station with more data that captures a range of years when comparing with a station with a different period of record that may not have as large a range or the same distribution. One way to account for this might be to allow the user to select the start year of the period of record in the web-tool in order to allow direct comparisons of the same years. The code used to generate the figures in this manuscript includes such functionality. We added a note in the methods that only sites with at least 20 years of data should be used, and added a note in the limitations that to facilitate comparisons between stations, similar periods of records should be used:**

**New text in methods:** "We recommend only using stations with at least 20 years of data."

**New text in limitations:** "First, we used the full period of record available for stations to calculate percentiles. In cases where stations being compared have sufficient data (i.e., at least 20 years) but differing periods of record, selecting commonly overlapping periods from which to calculate percentiles may avoid biases created by a station whose full record captures anomalous conditions (e.g., a notable wet, dry, warm, or cold set of years) compared to a station with a shorter period of record."

*Comment*

Line 420 – Could you say a little more specifically what concerns were highlighted in the papers you reference here?

**Certainly, thank you for the suggestion to be more specific. New text highlights a few specific concerns including runoff forecast timing, reservoir storage, and decreased water quality and quantity. New text:**

The goal of these diagrams and the web-based tool is to alleviate some management concerns outlined in Hossain et al. (2015) and Sterle et al. (2019), including runoff forecast timing errors, lack of upstream reservoir storage, and reductions in water supply reliability and water quality. These concerns can be start to be addressed through illuminating water supply uncertainties across a range of hydroclimate conditions, enhancing the flexibility of subseasonal-to-seasonal water management practices, and improving coupled atmospheric and hydrologic forecast systems (Siirila-Woodburn et al. 2021).

*Comment*

Conclusions – Could you say anything more about how the snow drought phase diagrams might support scientific innovation, rather than only focus on the management/end user audiences?

**Yes and thank you for pointing out the opportunity to broaden the application of the phase diagrams. We added a few examples to the conclusions:**

"Last, snow drought phase diagrams can support innovative ways in thinking about how weather and hydroclimate variability influence the mountain environment. For example, phase diagrams help us re-frame snow drought as a time-dependent process rather than a point-in-time concept, helping to contextualize hydrologic outcomes such as runoff efficiency. Phase diagrams also can be blended with other indicators relevant for ecosystem function (e.g., Contosta et al. 2019) to explore how snow drought impacts forest health or limnnology as well as changing wildfire behavior Alizadeh et al. (2021)."

**Added citations:**

Contosta, A. R., Casson, N. J., Garlick, S., Nelson, S. J., Ayres, M. P., Burakowski, E. A., Campbell, J., Creed, I., Eimers, C., Evans, C., Fernandez, I., Fuss, C., Huntington, T., Patel, K., Sanders-DeMott, R., Son, K., Templer, P., and Thornbrugh, C.. 2019. Northern forest winters have lost cold, snowy conditions that are important for ecosystems and human communities. Ecological Applications 29( 7):e01974. 10.1002/eap.1974

Alizadeh, M.R. Abatzoglou, J.T., Luce, C.H., Adamowski, J.F., Farid, A., Sadegh, M., (2021), Warming enabled upslope advance in western US forest fires, Proceedings of the National Academy of Sciences, 118 (22) e2009717118; DOI: 10.1073/pnas.2009717118

*Comment* I checked out the snow drought phase diagram tool – it was easy to use and is a great addition to the manuscript, in my opinion. My only critical comment on it was that I found that the color scale was not very intuitive (could you use the same color scale as Figure 2a?)

**Thank you for checking out the tool! We agree and are working with the web-developers to implement the same colormaps used in this manuscript to make for more intuitive, and color-safe, graphics on the web-tool. We appreciate you bringing this up, it helps the developers to learn about this feedback as well. They also appreciate the positive feedback about usability.**

**III. Technical corrections by line (L) number**

Line 26 – This sentence might read more easily if you started with "Reductions in snowpack negatively impact … in addition to …" Just a style suggestion; it was hard to see where it was going.

**We revised following this suggestion and agree the text reads better following this approach.**

Line 41 – typo, "hydroclimate conditions to varied .."

**Thank you, fixed.**

Line 71 - Should "Seaber" be inside parentheses?

**Thank you, fixed.**

Line 88 – I don't think 'drought-busting storms' should be in quotes unless you have a citation (even though I get your intention). I don't know if the meaning of "mixture effects" in this line is clear.

**Agreed, we added a citation to the quoted storm type (Dettinger, 2013). In the second sentence, we revised 'mixture effects' to be more specific to the time-varying behaviors of temperature and precipitation that ultimately control snowpack behavior. New text:**

Thus, phase diagrams can help diagnose snow drought onset, termination, duration, type, and severity as well as explore timing and characteristics of large 'drought-busting storms' (Dettinger, 2013). By implicitly including the time-varying effects of precipitation and temperature, phase diagrams provide a unique perspective over time series plots in tracking snowpack conditions.

**Added citation:**

Dettinger, M. D. (2013). Atmospheric Rivers as Drought Busters on the U.S. West Coast, Journal of Hydrometeorology, 14(6), 1721-1732. Retrieved Nov 12, 2021, from https://journals.ametsoc.org/view/journals/hydr/14/6/jhm-d-13-02_1.xml

Line 181 – A few times in this paragraph, you say snow or precipitation percentiles "improved" instead of "increased." While I agree with you, I think this inserts an unnecessary value judgement, and "increased" would probably be more clear.

**Agreed, thank you for pointing this out. All instances of 'improved' have been revised to 'increased'.**

Line 233-234: Using "active" to describe the weather feels a little imprecise; could this language be made more specific?

**Yes, we have replaced 'active' with the more descriptive (but still succinct) 'stormy'.**

Figure 7 – The color map here is clever and effective, but could you reverse the bar so that the lower numbers are on the left?

**Certainly! Thank you for the insightful suggestion to improve visual clarity. The revised manuscript includes the figure with a reversed colorbar.**

---

## Author Response (AR3)

**To:** Natural Hazards and Earth System Sciences
Dr. Veit Blauhut, Editor
**Re:** Technical Corrections to Accepted Manuscript (nhess-2021-193)
**Date:** 26-Jan-2022
* * *
Dear Dr. Blauhut,

Thank you for noting the issue with line 113 in the tracked changes version of the revised manuscript. The citations "Montecinos et al., 2017; Shortridge et al. 2019." overrunning the margin appears to be an issue with the diff function in LaTeX during the compiling process to create the tracked changes version. I was able to remedy this issue by forcing a newline (see Figure 1 below). The revised and clean (no tracked changes) manuscript looks good with no overrunning (see line 104; Figure 2).

Thank you again for your support throughout the submission and revision process, we greatly appreciate it. Sincerely,

Benjamin Hatchett

**3.1.1 Creating the Snow Drought Phase Diagram**

For each station, we calculated daily percentiles of accumulated precipitation and SWE from 1 November to 30 April (or 31 May) using a seven-day moving window centered on each calendar day to reduce seasonality effects (Montecinos et al., 2017; Shortridge et al., 2019). Results were not sensitive to moving windows sized between zero and 15 days. We calculated percentiles using the period of record. Following Huning and AghaKouchak (2020a), we used the U.S. Drought Monitor "D scale" (Svoboda et al., 2002) to characterize snow drought as abnormally dry (D0), moderate drought

Figure 1: Forced newline fixed the overrunning issue in the tracked changes version.

**3.1.1 Creating the Snow Drought Phase Diagram**

For each station, we calculated daily percentiles of accumulated precipitation and SWE from 1 November to 30 April (or 31 May) using a seven-day moving window centered on each calendar day to reduce seasonality effects (Montecinos et al., 2017; Shortridge et al., 2019). Results were not sensitive to moving windows sized between zero and 15 days. We calculated percentiles using the period of record. Following Huning and AghaKouchak (2020a), we used the U.S. Drought Monitor "D scale" (Svoboda et al., 2002) to characterize snow drought as abnormally dry (D0), moderate drought (D1), severe drought

Figure 2: Clean version of manuscript compiles correctly and has no overrunning.